# Re-evaluating Open-ended Evaluation of Large Language Models

**Siqi Liu\*, Ian Gemp\*, Luke Marris, Georgios Piliouras, Nicolas Heess, Marc Lanctot**
Google DeepMind
London, UK
`{liusiqi,imgemp,marris,gpil,heess,lanctot}@google.com`

## Abstract

Evaluation has traditionally focused on ranking candidates for a specific skill. Modern generalist models, such as Large Language Models (LLMs), decidedly outpace this paradigm. Open-ended evaluation systems, where candidate models are compared on user-submitted prompts, have emerged as a popular solution. Despite their many advantages, we show that the current Elo-based rating systems can be susceptible to and even reinforce biases in data, intentional or accidental, due to their sensitivity to redundancies. To address this issue, we propose evaluation as a 3-player game, and introduce novel game-theoretic solution concepts to ensure robustness to redundancy. We show that our method leads to intuitive ratings and provide insights into the competitive landscape of LLM development.

## 1 Introduction

We can only improve what we measure, yet measuring the performance of Large Language Models (LLMs) has become an elusive endeavor owing to their breadth and depth of capabilities. Real-world benchmarks are costly to curate, increasingly requiring feedback from human domain experts (Hendrycks et al., 2021; Rein et al., 2023). Synthetic benchmarks can help, but their relevance to real-world performance is less clear (Zhang et al., 2024; Hsieh et al., 2024). An even more vexing challenge of static benchmarks is that of test set contamination, a phenomenon difficult to prevent despite efforts (Golchin & Surdeanu, 2024; Balloccu et al., 2024; Palavalli et al., 2024). Enumerating skills of interests with narrowly defined static benchmarks seems to be an uphill battle from the outset, as frontier models become generally capable.

An emerging trend in LLM evaluation is therefore to rely on open-ended evaluation systems, a notable example being the LMSYS Chatbot Arena (Chiang et al., 2024). In such a system, users submit prompts of interest, with each model assigned an Elo score (Elo, 1978) based on how they compare to each other on all prompts. In contrast to static benchmarks, this open-ended approach enjoys liveness, diversity and scale, lending itself to become an important reference in LLM development. Despite an intuitive sense of progress, issues around redundancy, bias and quality of crowdsourced data have been raised (Chiang et al., 2024; Ahuja et al., 2023; Li et al., 2024b). Several recent studies reverted back to centralized curation for quality (Taori et al., 2023; Lee et al., 2024; White et al., 2024). Increasing commercial efforts have been invested in private and proprietary evaluation too.

Perhaps this tension between quality and open-endedness is to be expected in LLM evaluation. Biases, redundancies and quality issues in the prompt distribution can affect Elo ratings, as they reflect performance *on average*. This along with other identified deficiencies of the Elo system (Balduzzi et al., 2018; Bertrand et al., 2023; Lanctot et al., 2023) raise crucial questions for LLM development: how does an Elo-based open-ended evaluation system affect model development today, and how can we mitigate its drawbacks, if any, in the future? In this paper, we provide an empirical simulation-based investigation of the former and lean on game theory for a solution to the latter.

The connection between evaluation and game theory needs unpacking. Consider a set of agents and a set of tasks, a naive approach to evaluation would rank agents by their average performance over tasks, propagating biases and redundancies in the task set. A game-theoretic approach (Balduzzi

---

\*Equal contribution.

et al., 2018), would be to consider evaluation as an *agent-vs-task* game where the *agent* (*task*) player chooses one of its agents (tasks) and is rewarded (penalized) by the agent's performance on the task. This game-theoretic perspective accomplishes two goals simultaneously. First, it lets the evaluation system designers express their goals in players' objectives: here, Balduzzi et al. (2018) evaluates agents under *adversarial* task selection. Second, a game-theoretic equilibrium decides which actions are played during evaluation: quality and redundancies in players' action sets do not matter. It is in this sense that game theory is well suited for evaluation systems that are open-ended.

Applying game theory to LLM evaluation however has its own challenges. Indeed, the decision of Balduzzi et al. (2018) in comparing agents under *adversarial* task selection was guided by theoretical benefits. In 2-player zero-sum games, approximating a Nash equilibrium (NE, Nash et al. (1950)) is computationally tractable. NEs are also interchangeable in this setting as playing any NE guarantees zero exploitability. Beyond this setting, both benefits are lost: approximating NEs is computationally hard in the worst case (Daskalakis et al., 2006) and despite recent progress important challenges remain (Gemp et al., 2022; 2024). Equilibrium selection in this generalised setting remains a long-standing challenge too (Harsanyi & Selten, 1988; Rinott & Scarsini, 2000). For instance, driving on either side of the road is an equilibrium, but it is unclear which equilibrium should be used for evaluation. Past attempts at game-theoretic evaluation have therefore been restricted to the 2-player zero-sum settings when LLM evaluation calls for at least 3 players (e.g., *model-vs-model-vs-prompt*).

In this paper, we make several contributions that lead up to our equilibrium rating framework:

1. We show, via a simulated example (Section 1.1), the risk of models specializing in a few skills, at the expense of others, as they maximise their Elo ratings. Similarly, popular practice in prompt selection further reinforces this trend;

2. We introduce novel equilibrium solution concepts for $N$-player general-sum games that are unique and clone-invariant, a pre-requisite for our equilibrium rating method (Section 3);

3. We show our method scales to a real-world LLM evaluation dataset (Section 4.2) and provide ratings that are invariant to redundancy and correspond to our intuition in the sense of risk-dominance (Harsanyi & Selten, 1988), with empirical evidence (Appendix F.4);

4. We provide examples of analyzing these equilibrium structures of the game, drawing insights into the competitive landscape of LLM evaluation (Section 4.3).

## 1.1 ELO RATING IMPROVEMENT PATH: A SIMULATED EXAMPLE

With models continually improving their Elo ratings in systems such as LMSYS Chatbot Arena (Li et al., 2024a), it is worth asking if higher Elo scores translate to meaningful progress across skills of interest. This is difficult to answer from real-world data: we cannot replicate LLM development at scale nor can we disentangle factors driving model development besides maximizing leaderboard ratings. A synthetic example can provide insights in a controlled setting.

Consider $S$ orthogonal skills of interests, $M$ models and $P$ prompts with each prompt a probability vector $\boldsymbol{p} \in \Delta^S$ over the skills and each model a vector $\boldsymbol{m} \in \mathbb{R}_+^S$, representing its competencies in each skill. We can then define the utility of selecting model $\boldsymbol{m}_i$ when compared to model $\boldsymbol{m}_j$ on prompt $\boldsymbol{p}_k$, as $u_m(\boldsymbol{p}_k, \boldsymbol{m}_i, \boldsymbol{m}_j) = \boldsymbol{p}_k^T(\boldsymbol{m}_i - \boldsymbol{m}_j)$ with $i, j \in [M]$ and $k \in [K]$. A less common but equally valid question is what should be the utility, if any, for selecting a prompt. We follow a similar definition as Li et al. (2024b) and define the utility in choosing prompt $\boldsymbol{p}_k$ as $u_p(\boldsymbol{p}_k, \boldsymbol{m}_i, \boldsymbol{m}_j) = |u_m(\boldsymbol{p}_k, \boldsymbol{m}_i, \boldsymbol{m}_j)|$. The *separability* of a prompt is then $\frac{1}{M^2} \sum_{ij} u_p(\boldsymbol{p}_k, \boldsymbol{m}_i, \boldsymbol{m}_j)$, consistent with the prompt selection criterion used in offline benchmarks such as `arena-hard-v0.1`.

We now observe how this system evolves with rating-maximizing players. Consider two settings: a) the "initial prompts" setting where the set of prompts is fixed but the set of models expands; and b) the "additional prompts" setting where prompt and model players alternate to introduce new prompts and models. We use a simple evolutionary process for our simulation (see Appendix F.1 for pseudocode). Let $P_t$ and $M_t$ be the number of prompts and models at iteration $t$ and $P_0$, $M_0$ the number of initial prompts and models sampled from $\text{Dirichlet}(\mathbf{1}_{1:S})$. We introduce a model at each iteration which is a sum of improvement vectors sampled from $\text{Dirichlet}(\mathbf{1}_{1:S})$, such that the new addition receives the highest rating according the rating method used (i.e. Elo or our equilibrium-based method). In the "additional prompts" setting, a best-of-64 prompt is added at each iteration, selected by their separability when Elo ratings are used, and by their equilibrium ratings otherwise.

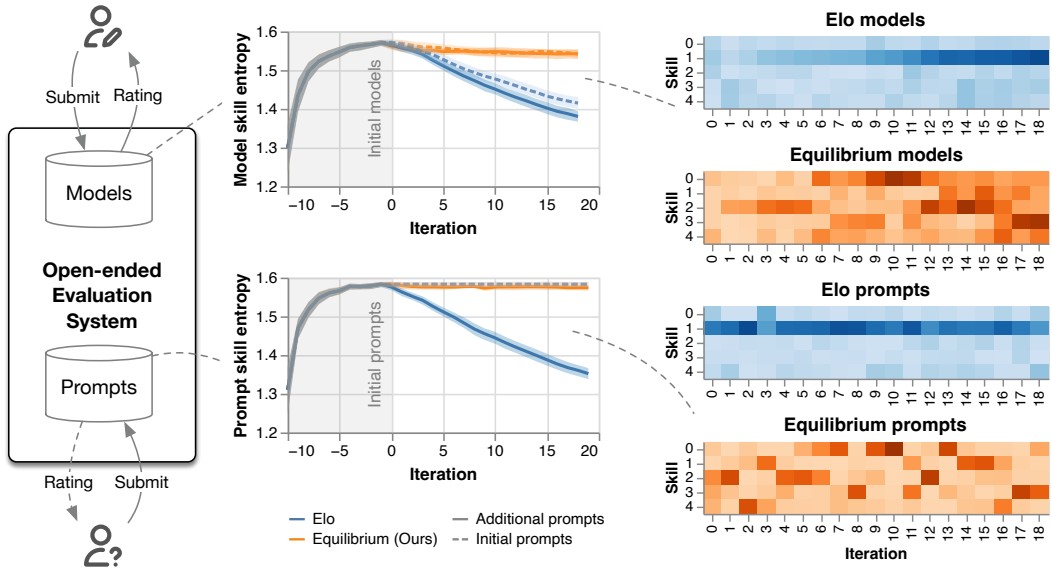

Figure 1: *(Left)* We simulate the effect of the rating method on model development with users submitting highly rated models (and prompts) iteratively. *(Center)* We show how model and prompt skill entropy evolves under different rating methods over 32 trials. *(Right)* We show an example sequence of models and prompts maximising their respective ratings. Darker indicates higher value.

Figure 1 *(Center)* shows our findings. Let $H(\bar{\boldsymbol{p}}_t)$, $H(\bar{\boldsymbol{m}}_t)$ be the prompt and model skill entropy at iteration $t$ with $\bar{\boldsymbol{p}}_t = \frac{1}{P_t} \sum_i^{P_t} \boldsymbol{p}_i$ and $\bar{\boldsymbol{m}}_t = \frac{1}{M_t} \sum_i^{M_t} \boldsymbol{m}_i$ and $H$ the Shannon entropy. The Elo rating method leads to a consistent decline in skill entropy: the sequence of models improve along specific skill dimensions that are over-represented in the fixed set of initial prompts (dashed). Adding prompts with high separability further reinforces this trend in both model and prompt skill entropy (solid). We offer an intuitive explanation. Improvement on the Elo ratings or the separability metric reflects improvements against the *average*. At iteration $t$, the expected utility to model $\boldsymbol{m}_i$ is given by $u_m(\bar{\boldsymbol{p}}_t, \boldsymbol{m}_i, \bar{\boldsymbol{m}}_t)$ with its gradient defined by $\bar{\boldsymbol{p}}_t$ — improving on the most prevalent skill in $\bar{\boldsymbol{p}}_t$ therefore leads to the steepest ascent in model utility. Similarly, the gradient for a prompt vector $\boldsymbol{p}_k$ is defined by the absolute deviation of the model vectors along each skill dimension $\frac{1}{M_t} \sum_i^t |\boldsymbol{m}_i - \bar{\boldsymbol{m}}_t|$. Prompts that target the skill dimension with the highest "spread" averaged across all model pairs are therefore the most highly rated. Figure 1 *(Right)* illustrates this phenomenon from a single trial. The Elo-maximising sequence of models specialises in skill 1 due to prompt redundancy while the separability-maximising sequence of prompts remains focused on skill 1, at the expense of others.

The underlying challenge, one that we address, is to propose a practical rating method that compares models and prompts in a way that is intuitive and robust to redundancies. Figure 1 *(Center)* suggests that maximising equilibrium ratings preserves skill entropy. Indeed, Figure 1 *(Right)* shows models focusing on different skills across iterations. As prompts are no longer measured against the *average* model pairs, they also remain diverse. In both cases, ratings are computed at a game-theoretic equilibrium distribution, instead of the uniform. We now present our equilibrium rating framework.

## 2 BACKGROUND

**Normal-form game** A normal-form game is a tuple $(N, \mathcal{A}, u)$ where $N$ is a finite set of players $N = \{1, \ldots, n\}$ indexed by $i$, a tuple of strategy (action) sets $\mathcal{A} = (\mathcal{A}_1, \ldots, \mathcal{A}_n)$, and a tuple of utility functions $u = (u_1, \ldots, u_n)$ with $\mathcal{A}_i$ and $u_i : \mathcal{A} \to \mathbb{R}$ being player $i$'s strategy set and utility respectively. Let $\boldsymbol{a} \in \mathcal{A} = (a_1, \ldots, a_n)$ with $a_i \in \mathcal{A}_i$ for all $i$ denote a strategy *profile*. We allow strategy profiles to be selected randomly according to a distribution $\boldsymbol{x} \in \Delta(\mathcal{A})$ over *joint* actions. Let $x_i$ denote the marginal distribution over player $i$'s strategy set $\mathcal{A}_i$, i.e., $\boldsymbol{x}$ with all players $j \neq i$

marginalized out. Likewise, let $x_{-i}$ denote the distribution with player $i$ marginalized out. We call $\boldsymbol{x}$ a *pure* strategy if it places all mass on a single action profile and *mixed* otherwise. Each player's utility function is naturally extended to randomized strategy profiles by considering its expected value $u_i(\boldsymbol{x}) = \mathbb{E}_{\boldsymbol{a} \sim \boldsymbol{x}}[u_i(\boldsymbol{a})]$. Similarly, let $u_i(x_i, x_{-i}) = \mathbb{E}_{\substack{a_i \sim x_i \\ a_{-i} \sim x_{-i}}}[u_i(\boldsymbol{a})]$.

**Coarse Correlated Equilibrium (CCE) and Nash Equilibrium (NE)**  An equilibrium is a strategy profile $\boldsymbol{x}$ from which no player has an incentive to unilaterally deviate. Define player $i$'s incentive to deviate to $x_i' \in \Delta(\mathcal{A}_i)$ unilaterally as $\text{regret}_i(x_i', \boldsymbol{x}) = u_i(x_i', x_{-i}) - u_i(\boldsymbol{x})$, where $\Delta(\mathcal{A}_i)$ is the simplex over $\mathcal{A}_i$. Then, player $i$'s maximum regret for deviating from $\boldsymbol{x}$, is defined as:

$$\max_{x_i' \in \Delta(\mathcal{A}_i)} \Big[ \text{regret}_i(x_i', \boldsymbol{x}) \Big] = \max_{x_i' \in \Delta(\mathcal{A}_i)} \Big[ u_i(x_i', x_{-i}) \Big] - u_i(\boldsymbol{x}). \tag{1}$$

The profile $\boldsymbol{x}$ is an approximate Coarse Correlated Equilibrium (Aumann, 1974; 1987) ($\epsilon$-CCE) iff $\forall i, \max_{x_i' \in \Delta(\mathcal{A}_i)} \Big[ \text{regret}_i(x_i', \boldsymbol{x}) \Big] \leqslant \epsilon$. If $\boldsymbol{x}$ can be factorized into player marginals such that players cannot correlate, i.e., $\boldsymbol{x} = \times_{i=1}^{N} x_i$, then $\boldsymbol{x}$ is also an $\epsilon$-NE. NEs are a subset of CCEs.

**Equilibrium Selection**  Games can have many equilibria. Additional criteria are often introduced to make their selection unique. The set of CCEs is always convex, and so any strictly convex objective function such as negative Shannon entropy can used to select a unique equilibrium.

In contrast, the set of NEs need not be convex, however, several solutions have been proposed to solve for unique Nash equilibria in general-sum games (Harsanyi & Selten, 1988). The LLE was originally defined by McKelvey & Palfrey (1995) along with their introduction of quantal response (logit) equilibria (QREs) which satisfy the following fixed point equation for all players $i \in N$:

$$x_i = \text{softmax}(\frac{1}{\tau} \nabla_{x_i} u_i) \tag{2}$$

where $\nabla_{x_i} u_i$ is the gradient of $u_i$ w.r.t. $x_i$. QREs are defined by a temperature parameter $\tau$ and can be interpreted as the Nash equilibria of a game with payoffs perturbed by $\text{Gumbel}(0, \tau)$ noise. Computing the LLE involves tracing a continuum of QREs, starting at temperature $\tau = \infty$ (corresponding to the uniform strategy profile) and ending at the LLE in the limit of $\tau = 0$. The LLE is unique in all games except a 0-measure set (McKelvey & Palfrey, 1995; Goeree et al., 2003). Another reason to solve for an LLE is that it falls into the family of homotopy methods (Herings & Peeters, 2010), which were shown to select *risk-dominant* equilibria in some general settings, a Nobel prize winning result of Harsanyi & Selten (1988). Empirically, LLEs have also been shown to approximate human play in games (McKelvey & Palfrey, 1995; Goeree et al., 2003).

## 3  METHOD

We now describe our rating method in terms of gamification, equilibrium solving and its selection. In gamification, we endow prompt and model players with utility functions, partly inspired by prior works, such that actions played at an equilibrium reflect our intuition. We note that our specific gamification defines an $N$-player general-sum game where equilibrium solving and selection requires more careful consideration. For equilibrium solving, we build on existing methods for approximating NEs and CCEs, reformulated to accommodate entropy-based techniques that select unique equilibria and explain why ratings derived from these equilibria remain vulnerable to manipulation in the face of redundant actions. We then propose a family of algorithms based on a novel kernelized entropy that select unique equilibria yet are also robust to redundant actions. Finally, for a given equilibrium solution $\boldsymbol{x}$, we define the rating of an action $a_i$ to be $\text{regret}_i(a_i, \boldsymbol{x})$.

### 3.1  GAMIFICATION: EVALUATION VIA A GAME BETWEEN MODELS AND PROMPTS

We study a 3-player general-sum game in our experiments. Consider a *prompt* player with $a_p \in \mathcal{A}_p$ the set of prompts, a *king* player and a *rebel* player each with actions $a_m \in \mathcal{A}_m$ the set of models. Let $u_k(a_p, a_m, a_m') \in \{-1, -1/2, 0, +1/2, +1\}$ be the utility function to the king player representing a preference towards king model response $a_m$ over the rebel model response $a_m'$ on a prompt $a_p$.

The prompt player is rewarded for separating the models, with $u_p(a_p, a_m, a'_m) = |u_k(a_p, a_m, a'_m)|$. The rebel player receives $u_r(a_p, a_m, a'_m) = -u_k(a_p, a_m, a'_m)$ except for when $a_m = a'_m$ in which case $u_r(a_p, a_m, a'_m) = -1$. This asymmetry discourages the same model being played by both model players deterministically with a prompt player indifferent over its actions. We refer to this game as king-of-the-hill as it favours the king player, leaving the rebel player to mount its best resistance without relying on some of the best models that the king player may choose. We refer to the king player ratings as the model ratings in our results.

Given a collection of prompts and models, the utility function can be tabulated with $|\mathcal{A}_p| \times |\mathcal{A}_m|^2$ pairwise preference ratings. We query a gemini-1.5-pro-api-0514 judge for preference ratings similar to Zheng et al. (2023); Verga et al. (2024); Dubois et al. (2024a;b); Chiang & Lee (2023); Liu et al. (2023). We caveat that our results could therefore suffer from *self-preference* (Panickssery et al., 2024) and should *not* be viewed as an objective assessment of frontier LLMs.

## 3.2 EQUILIBRIUM SOLVING

For an instance of the evaluation game, we can compute different equilibrium solutions $x$ which then define ratings. Here we present two options as they are unique, scalable and lead to intuitive, invariant ratings when combined with a selection criteria that we describe in Section 3.3.

**Nash Equilibrium (NE)** While LLE computation (Turocy, 2005) is typically formulated as solving a differential equation that evolves the temperature $\tau$ towards 0 while obeying the logit constraint in Equation (2), i.e., $x_i = \text{softmax}(\frac{1}{\tau}\nabla_{x_i} u_i)$ for all $i$, this is also equivalent to satisfying the constraint $x_i = \arg\max_{z_i \in \Delta} u_i(z_i, x_{-i}) + \tau S(x_i)$ where $S(x_i)$ is the Shannon entropy of $x_i$. In this work, we choose another condition

$$x_i = \arg\max_{z_i \in \Delta} \left\{ u_i^\tau(z_i, x_{-i}) \overset{\text{def}}{=} u_i(z_i, x_{-i}) - \tau D_{\text{KL}}(z_i \| t_i) \right\} \tag{3}$$

which is equivalent in the case where the target strategy $t_i$ is set to player $i$'s uniform strategy. Using this definition of $u_i^\tau(z_i, x_{-i})$, we can define a loss function as in (Gemp et al., 2022) such that $\arg\min_x \mathcal{L}^\tau(x)$ is a QRE at temperature $\tau$:

$$\mathcal{L}^\tau(x) = \sum_i u_i^\tau(\text{BR}_i, x_{-i}) - u_i^\tau(x_i, x_{-i}) \tag{4}$$

where player $i$'s best response $\text{BR}_i = \text{softmax}(\frac{1}{\tau}\nabla_{x_i} u_i + \log(t_i))$. By annealing $\tau$ from a high value and successively re-solving for the global minimum of $\mathcal{L}^\tau$, we can approximately trace the QRE continuum to the LLE. In Section 3.3, we explore non-uniform $t_i$ to achieve clone-invariance.

**Coarse Correlated Equilibrium (CCE)** Solving for a unique CCE is computationally easier than NE as the problem is convex (Equation (1)). Therefore any strictly convex function can be used to uniquely select an equilibrium. For example, maximum entropy would be a suitable default criterion following the principle of maximum entropy. However, as we show in Section 3.3, a different target formulation is necessary for clone-invariance. As such, we opt for maximum relative entropy to a target joint $t = \times_{i=1}^n t_i$ to allow for non-uniform target joint distributions. A number of off-the-shelf solvers (Domahidi et al., 2013) and frameworks (Diamond & Boyd, 2016) can be used to compute solutions to this problem. We used a particularly efficient dual space gradient based algorithm described in Appendix A for scaling.

## 3.3 INVARIANT EQUILIBRIUM SELECTION

There may be many NEs and CCEs (McLennan & Park, 1999; Sturmfels, 2002; McLennan, 2005). Some equilibria exhibit sparse or heavily skewed strategy profiles (see examples in Appendix F.4). Intuitively, these equilibria are *risky* in the sense of *risk dominance*: playing one such equilibrium when other players do not would be a costly mistake. Our goal is to propose a selection procedure that along with our equilibrium solving algorithms, approximates a clone-invariant equilibrium.

Shannon entropy plays a key role in several equilibrium selection approaches, however, its definition is vulnerable to redundancy in games. Consider a game with 2 distinct actions $A$ and $B$ per player and introduce $b-1$ clones of $B$ into player 1's action set. The maximum entropy strategy for player 1

in the new game is uniform across their actions with mass $\frac{1}{1+b}$ on each, but this induces a distribution that places $\frac{b}{1+b}$ cumulative mass on the cloned action $B$. From Section 3.2, the maximum Shannon entropy profile defines the precise starting point for tracing the path of QREs towards the LLE. This starting point is sensitive to clones. Hence, if we compute the LLE using the uniform distribution in this new game, we will effectively start from the $(A, B)$ mixed-strategy $(\frac{1}{1+b}, \frac{b}{1+b})$ rather than the desired mixed-strategy $(1/2, 1/2)$; hence, will not necessarily arrive at the LLE of the original game.

**Desired properties.** A clone-invariant entropy definition should be:

**P1.** Real-valued, finite, and non-negative for any distribution $x$;

**P2.** Have a well-defined gradient for any $x$ in the interior of the simplex;

**P3.** Its maximizers should form a convex set. In the case of duplicate strategies (clones), the maximizers should form precisely the set of distributions which arbitrarily distribute a mass of $\frac{1}{c}$ across each of the $c$ sets of clones. In addition, they should achieve an entropy value which is equal to the entropy of the system with clones removed;

**P4.** Amenable to efficient estimation and flexible to re-interpretation of redundancy.

Note **P3.** resolves the issue with Shannon entropy that we highlighted above. **P1** is necessary for a reasonable measure of information content. **P2** is necessary for gradient-based optimization, and **P4** is practically helpful for efficient implementation and adaptation to bespoke game settings. We now introduce *affinity* entropy $H_a^p : \Delta \to \mathbb{R}$, a generalized Tsallis entropy (Tsallis, 1988) that recognises similar or redundant strategies. Its derivation from the above axioms can be found in Appendix B.

**Definition 1** (Affinity Entropy $H_a^p$).

$$H_a^p(\boldsymbol{x}) = \frac{1}{p}\Big[1 - \mathbf{1}^\top (U^{(p)}\boldsymbol{x})^{p+1}\Big] \tag{5}$$

with entropic-index parameter $p \in (0, 1]$, $U^{(p)} = K\Lambda_p^{-1}$, and $K$ a similarity kernel with entries in $[0, 1]$ with 1 indicating two strategies are clones, and $\Lambda_p$ a diagonal matrix containing the $(p+1)$-norms of the columns of $K$ on its diagonal.

**Theorem 1.** *Affinity entropy $H_a^p$ satisfies all desiderata **P1-P4**.*

In experiments, we define a similarity kernel $K^{(i)}$ for each player $i$ with entries $K_{\alpha\beta}^{(i)}$ with

$$D_{\alpha\beta}^{(i)} = \mathbb{E}_{\boldsymbol{a} \sim U(\mathcal{A})}\big[\big(u_i(\alpha, a_{-i}) - u_i(\beta, a_{-i})\big)^2\big] \tag{6}$$

$$K_{\alpha\beta}^{(i)} = \exp(-D_{\alpha\beta}^{(i)}/(2\sigma)^2) \tag{7}$$

where $D$ measures the strategic *dis*-similarity between player $i$'s strategies $\alpha$ and $\beta$ and $K$ is simply a radial basis function (RBF) kernel under the metric $D$. Note $D_{\alpha\beta}^{(i)}$ is zero iff two strategies $\alpha$ and $\beta$ achieve exactly the same utility for player $i$ irrespective of the actions chosen by other players in the game. It should also be clear from the definition how one might Monte-Carlo estimate $D$. To select for an NE or a CCE, we set $t = \arg\max H_a^{p=1}(x)$ in Equation (3) and Equation (8) respectively.

## 4 RESULTS

We use the same hyper-parameters for equilibrium solving in all results (see Appendix F.2). For evaluation on real-world prompts, we consider the `arena-hard-v0.1` dataset with 500 prompts, selected to separate frontier LLMs, as well as responses from many candidate LLMs. We consider responses from 17 LLMs in particular and queried `gemini-1.5-pro-api-0514` for 8 pairwise preference ratings on each prompt for each model pair. See Appendix F.3 for more details.

### 4.1 EQUILIBRIUM RATING IMPROVEMENT PATH: A SIMULATED EXAMPLE

Recall from Figure 1 that contrary to the Elo improvement path, maximizing equilibrium ratings led to models (and prompts) improving across skills. We inspect the equilibrium improvement path and offer our interpretation. Figure 2 *(Right)* shows that the shifts in focus between skills by the model

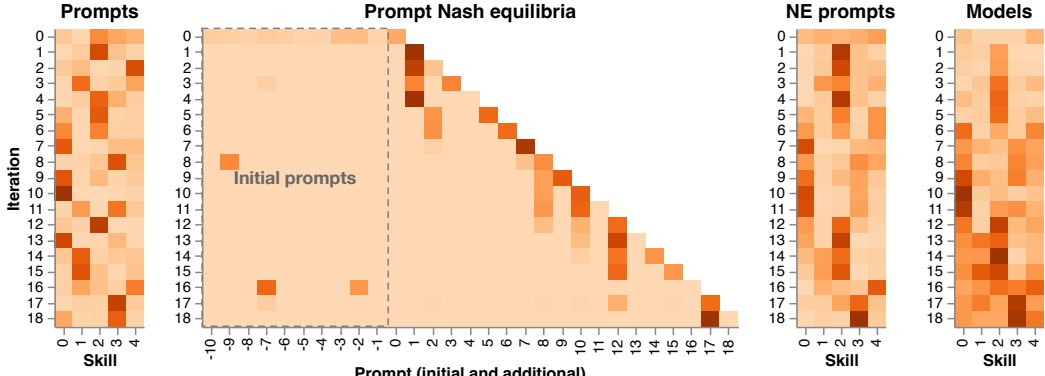

Figure 2: We inspect the model improvement path induced by NE ratings as shown in Figure 1 *(Right)*. *(Left)* shows the sequence of additional prompts added at each iteration. Each prompt is the best-of-64 samples according to their NE ratings. *(Center)* shows the sequence of prompt player NEs. Each row defines a distribution over prompts. *(Right)* shows the equilibrium-weighted prompt skills and the sequence of king player models. Recall prompts and models are non-negative vectors over skills, darker indicates higher focus or capability in each skill.

player coincides with transitions in the NE prompts, or prompts weighted by their NE strategies (shown in Figure 2 *(Center)*). Similarly, to gain support under an NE, new prompts must highlight a skill dimension along which equilibrium models are better differentiated (Figure 2 *(Left)*). In sum, equilibrium prompts separate equilibrium models. This dynamic encourages exploration of new skill dimensions and incentivises models to be well-rounded across skills.

## 4.2 INVARIANT EVALUATION

We now turn to `arena-hard-v0.1` and show that candidate LLMs' equilibrium ratings are invariant to redundancies when their Elo ratings are not. In this experiment, we will introduce prompts targeted at bringing down the rating of a certain action (in this case, model). Specifically, let $\bar{\boldsymbol{u}}_k(a_k) = \frac{1}{|\mathcal{A}_m|} \sum_{a_r} u_k(\cdot, a_k, a_r)$ be the vector of expected king player payoffs when playing action $a_k$ against a randomly chosen rebel model on each prompt. We can then sample prompts adversarial to $a_k$ from $\mathrm{softmax}(-\lambda \bar{\boldsymbol{u}}_k(a_k))$ and add them to the prompt set. Figure 3 reports the king model rankings under different methods with $a_k = $ `gemini-1.5-pro-api-0514` and $\lambda = 10$.

Our first observation is that without redundant adversarial prompts, our proposed equilibrium rankings of LLMs are fairly consistent with their Elo rankings, with a few models moving up or down one or two positions. This deserves attention. Out of a multiplicity of equilibria, the NE and CCE we selected led to rankings that correspond to our intuition. Indeed, we show in Appendix F.4 that the NE we select is risk-dominant among 128 mixed-strategy NEs of this game. Second, the Elo ratings can be arbitrarily influenced by redundancy, with the top-ranked model falling through the ranks. Equilibrium rankings remain invariant. In fact, while we lose the invariance guarantee with *near* redundant prompts, we show models' equilibrium rankings to degrade gracefully in Appendix F.5. Third, the CCE ratings show the top-3 models to tie for the first place: correlating models with prompts affects the competitive landscape which we inspect in Section 4.3. Lastly, solving for a unique equilibrium is not sufficient for invariant ratings. We show in Figure 3 *(Right)* that using Shannon's entropy for tracing the QRE continuum or for selecting a max-entropy CCE would not lead to invariant ratings. For completeness, we provide a detailed breakdown of our equilibrium ratings in terms of action ratings and marginals for each player in Appendix F.5.

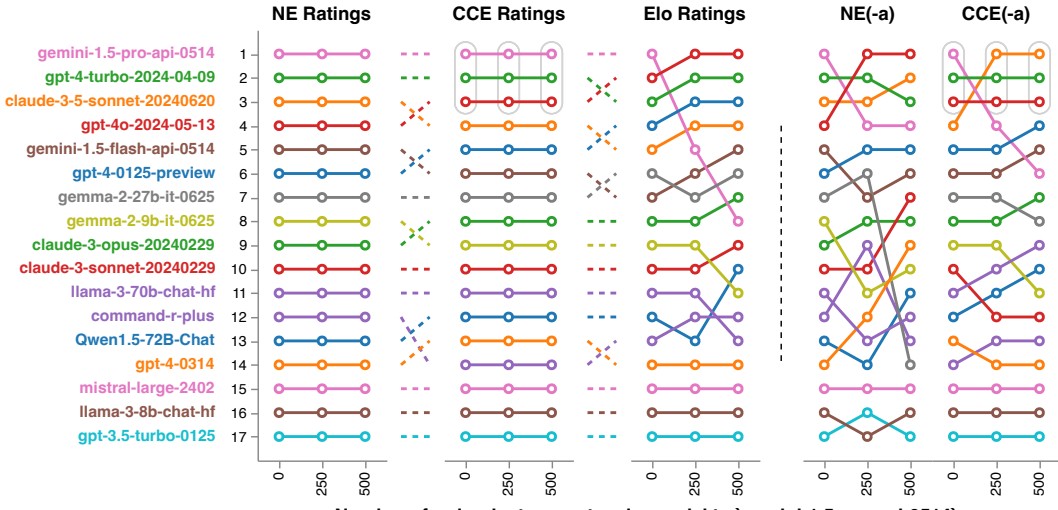

Figure 3: We introduce an increasing number of redundant copies of prompts adversarial to `gemini-1.5-pro-api-0514` and show model rankings under each method. Models at the same rank are grouped in grey and ordered alphabetically. *(Right)* We show equilibrium rankings under NE(-a) and CCE(-a) selected using Shannon's entropy instead of the affinity entropy. Dotted lines connecting different rating panels indicate continuity in the labeling. For instance, `gemini-1.5-pro-api-0514` consistently ranks first under our NE and CCE ratings, despite the introduction of up to 500 redundant adversarial prompts. However, its ranking suffered significantly under the Elo ratings as soon as 250 adversarial prompts have been introduced.

## 4.3 INTERPRETING EQUILIBRIUM SOLUTIONS

Besides rankings, the equilibrium solutions can surface interpretable insights. We share two examples using NE and CCE solutions respectively from ratings shown in Section 4.2.

**Nash Equilibrium Prompts** We have shown that equilibrium ratings are intuitive and invariant to redundancy. A follow-up question is which actions are highly-rated and which actions affect other players' ratings (i.e., with positive support at the NE).

Recall that the prompt player utility $u_p(a_p, a_k, a_r) = |u_k(a_p, a_k, a_r)|$ reflects the extent to which a prompt separates the pair of responses from models $a_k$ and $a_r$. The prompt player's equilibrium rating is then $\text{regret}(a_p, \boldsymbol{x}) = \mathbb{E}_{\substack{a_k \sim x_k \\ a_r \sim x_r}} u_p(a_p, a_k, a_r)$ with $x_k$, $x_r$ the NE strategies of the king and rebel player respectively. By definition, prompts that are highly rated under NE ratings separate models played at the NE. In other words, while the Elo ratings reflect the strength of an action on average, equilibrium ratings reflect the strength of actions at the selected equilibrium.

We can now illustrate these phenomena using the same game investigated in the second columns of Figure 3, with 250 redundant prompts added to the game. First, we show in Figure 4 *(Top)* the *king-vs-rebel* payoff matrices induced by 6 sample prompts, with increasing equilibrium prompt ratings. Prompts with low ratings tend to fail to differentiate performant models (i.e. top-left block of each heatmap). Second, we can ask which prompts should we expect to have support at an equilibrium. Figure 4 *(Bottom)* shows that empirically, highly rated prompts are played more often at the equilibrium we select. This implies that the model ratings are heavily influenced by a small subset of prompts that separate frontier models. We note that this correlation is not guaranteed, following our discussion in Section 4.2 on redundant actions. Indeed, our final observation is that prompts that are clones with other prompts tend to receive lower probability mass than their ratings would have required. In fact, since we have introduced 250 redundant prompts explicitly, we can highlight in gray prompts that are indeed redundant — many of these prompts enjoy high ratings, but significantly lower mass. In other words, equilibrium ratings reflect quality of an action in isolation

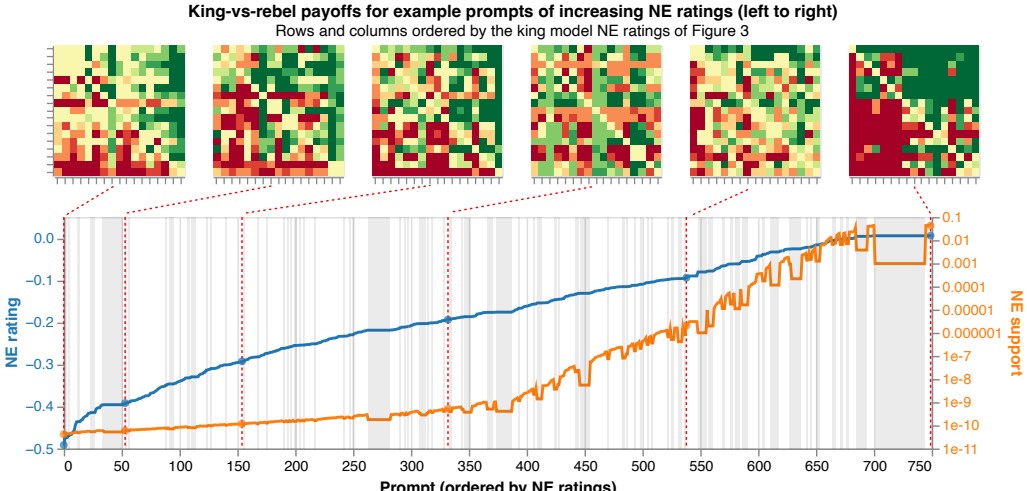

Figure 4: Highly rated prompts generally have high support under the NE. Redundant prompts (gray bands) receive identical ratings but notably lower support. In sum, equilibrium ratings reflect separability of each prompt with respect to the model equilibrium strategies in isolation, whereas equilibrium support of each prompt further accounts for its redundancy with respect to other prompts. *(Top)* We show the *king-vs-rebel* payoffs induced by example prompts. Green indicates king-player winning and red losing. Highly rated prompts tend to discriminate between strong models (top-left corners). *(Bottom)* We show the NE supports and ratings of all prompts, ordered by their NE ratings.

while equilibrium mass further takes into account redundancy of an action with respect to other actions. This observation is even clearer in games studied in Appendix D-E.

**Marginal rating contribution by co-player action** With ratings derived from underlying equilibria, we can decompose the rating of each action into a sum of marginal contributions from each co-player's actions. Recall from Equation (1) that the rating of an action $a_i'$ is its $\mathrm{regret}_i(a_i', \boldsymbol{x}) = \sum_a x(a) \left[ u_i(a_i', a_{-i}) - u_i(a) \right]$. We can decompose the rating of player $i$'s action $a_i'$ into a weighted sum of each of player $j$'s contributions, with $\delta(a_i', a_j, \boldsymbol{x}) = \sum_{a_{-j}} x(a) \left[ u_i(a_i', a_j, a_{-i,-j}) - u_i(a_j, a_{-j}) \right]$ the marginal contribution of $a_j$ to $a_i'$'s equilibrium rating. Note that $\mathrm{regret}(a_i', \boldsymbol{x}) = \sum_{a_j} \delta(a_i', a_j, \boldsymbol{x})$. The marginal contribution $\delta(a_i', a_j, \boldsymbol{x})$ therefore explains $a_j$'s contribution to player $i$'s decision to not deviate.

Recall from Figure 3 where several models tied for the first place under the CCE profile but are fully differentiated under NE. We can now leverage the marginal contribution analysis to understand the mechanism underlying this phenomenon. Figure 5 shows the CCE king model ratings decomposed from the perspective of the rebel player. In other words, we ask which rebel models contribute most positively or negatively to each king model's CCE rating. For clarity of presentation, we focus on the top 5 models and we group rebel models into families of models if they share the same naming prefix. The contribution of each family of model is therefore the sum of the contribution by models within each family $\mathcal{F}$ or $\sum_{a_r \in \mathcal{F}} \delta(a_k', a_r, \boldsymbol{x})$ with $a_k'$ a king model and $a_r$ a rebel model.

We make several remarks. First, all 3 top-ranked king models benefit the most when compared against rebel models in their own model family: the GPT family (Achiam et al., 2023) of models contribute positively to the ratings of gpt-4o-2024-05-13 and gpt-4-turbo-2024-04-09. Similarly, gemini-1.5-flash-api-0514, the only other model in the Gemini family (Team et al., 2023), improves gemini-1.5-pro-api-0514's rating the most. We speculate that this can be a result of model developers selecting models to release based on favourable comparisons to their earlier or smaller models. Second, all top-ranked models remain vulnerable to open-weight models such as the Mistral (Jiang et al., 2023) and Llama (Dubey et al., 2024) families of models. More fine-grained analysis may shed light on the prompts on which these losses tend to occur.

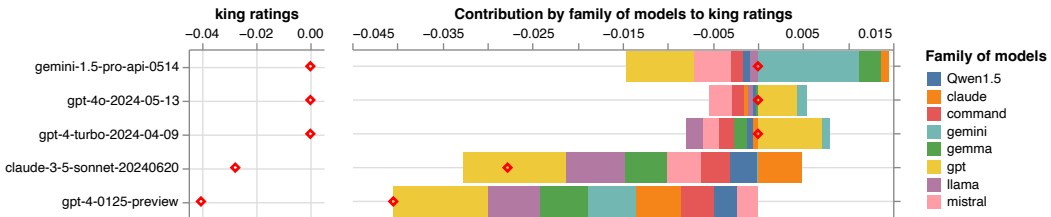

Figure 5: The CCE joint distribution can surface insights in the comparison data. Each bar represents a model family $\mathcal{F}$ and its width corresponds to $\sum_{a_r \in \mathcal{F}} \delta(a'_k, a_r, \boldsymbol{x})$ with $a'_k$ a king player model choice and $a_r$ a rebel model belonging to the family $\mathcal{F}$. A model's family is determined by its model name prefix. For brevity, we show the king model rating breakdown for the top 5 models.

We caveat that our results are in part derived from the preference ratings of a `gemini-1.5-pro-api-0514` model and may not reflect the true dynamics of real-world LLM development. Nevertheless, the interpretability offered by the game-theoretic equilibria further distinguishes game-theoretic evaluation from prior works to be discussed in the Section 5.

## 5 RELATED WORKS

There is a rich body of literature studying rating methods with applications in Chess, Go, Tennis and video games. One family of probabilistic methods follows the Bradley-Terry model and predicts pairwise win probabilities from ratings. A widely used example is Elo (Elo, 1978) with extensions Bayes-Elo, mElo and Elo-MMR (Coulom, 2008; Balduzzi et al., 2018; Ebtekar & Liu, 2021; Vadori & Savani, 2024) capturing temporal variation, cyclicality and ordinal ranks in data. Elo ratings can typically be efficiently solved as regression problems, although their ratings are vulnerable to redundancy. A separate line of work draws from Social Choice (or *Voting*) Theory (SCT, Sen (1977); Lanctot et al. (2023)), which also studies *independence of clones*: rankings should be invariant to redundant candidates (e.g., LLM models) being added. However, invariance to redundancy in votes (e.g., prompts) is in direct opposition to the spirit of social choice theory. In this sense, SCT provides partial (one-sided) clone invariance, which we argue is insufficient for open-ended, LMSYS-style evaluation. Finally, game-theoretic evaluation has been previously studied in Balduzzi et al. (2018) and Marris et al. (2022b) where full clone-invariance is guaranteed in the 2-player zero-sum setting. Our method generalises the approaches in these works to $N$-player general-sum settings, with practical equilibrium solving and selection algorithms based on our novel affinity entropy definition. Other approaches have been concurrently developed that avoid the equilibrium selection dilemma, and hence obviate the use of entropy (Marris et al., 2025).

## 6 CONCLUSIONS

We studied the effect of maximizing Elo ratings in the context of open-ended evaluation and showed that its sensitivity to redundancy could bias model (and prompt) selection. We then proposed an equilibrium rating framework, with practical equilibrium solving and selection algorithms that can scale to real-world LLM evaluation. We show our method to provide intuitive and robust rankings of models (and prompts), with interpretable structures.

We see several exciting future directions. First, although our methods can scale to tens of thousands of prompts and tens of models on commodity hardware, scaling further would be challenging. Tabulating the evaluation payoff tensor with pairwise preference ratings can be costly too. Research into alternative solution concepts, or how we could leverage their equilibrium structure for analysis (e.g. prompt and model pruning) is also promising. Finally, while we target LLM evaluation in particular, our methodology can be applied more generally to other domains. For instance, our rating methods could evaluate multi-modal model generation capabilities (Jiang et al., 2024) or analysing game dynamics for video game development (Pendurkar & Chow, 2023).

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

## A  COMPUTING THE MAXIMUM RELATIVE ENTROPY CCE

A maximum relative entropy CCE, that minimises the distance of the log-joint, $\log(x(a))$, to a target log-joint, $t(a) \in \mathbb{R}^{|\mathcal{A}|}$, can be computed using gradient descent. We formulate the problem in dual space (Marris et al., 2022a) with dual parameters, $\alpha_i(a_i') \in \mathbb{R}_+^{|\mathcal{A}_i|} \; \forall i$, defined as functions, $\alpha_i(a_i') = \text{softplus}(\theta_i(a_i')) \; \forall i$, of learned parameters, $\theta(a_i') \in \mathbb{R}^{|\mathcal{A}_i|} \; \forall i$. Let $l_\theta(a)$ be a logit term used to construct the loss function.

$$l_\theta(a) = -\sum_i \sum_{a_i'} \alpha_p(a_i') \left[ u_i(a_i', a_{-i}) - u_i(a) \right] + t(a) \tag{8}$$

Minimizing a loss function, $\min_\theta L_\theta$, converges to optimal dual variables, $\alpha_i^*(a_i') = \text{softplus}(\theta_i^*(a_i')) \; \forall i$ with $L_\theta = \log \left[ \sum_a \exp \left[ l_\theta(a) \right] \right]$. The loss is convex, deterministic, and unconstrained. Therefore many optimization algorithms are suitable. The primal joint can be simply recovered from the optimal logit term $x_\theta(a) = \text{softmax} \left[ l_{\theta*}(a) \right]$.

## B  AFFINITY ENTROPY

Consider defining a modified Tsallis entropy $H_a^p$ with temperature parameter $p \in (0, 1]$ as:

$$H_a^p(\boldsymbol{x}) = \frac{1}{p} \left[ 1 - \boldsymbol{z}^\top \boldsymbol{z} \right] = \frac{1}{p} \left[ 1 - \sum_i (U_i^{(p)} \boldsymbol{x})^{p+1} \right] \tag{9}$$

where $\boldsymbol{z} = (U^{(p)} \boldsymbol{x})^{\frac{p+1}{2}}$. Note that this definition recovers the standard definition of Tsallis entropy when $U^{(p)}$ is the identity matrix.

**Remark.** $U_{ij}^{(p)} \geqslant 0$ *for all entries for $H_a^p$ to be real-valued.*

$U_{ij}^{(p)}$ must be non-negative for every $i, j$, otherwise, there exists $\boldsymbol{x} = \boldsymbol{e}_j$ where $\boldsymbol{e}_j$ is a standard-basis vector such that $U_i^{(p)} \boldsymbol{x} < 0$ and $(U_i^{(p)} \boldsymbol{x})^{p+1}$ is not real for $p \in (0, 1)$.

**Remark.** *The $(p + 1)$-norm of each column of $U^{(p)}$ must be less than or equal to $1$ for $H_a^p$ to be non-negative for any $\boldsymbol{x} \in \Delta$.*

We need $\boldsymbol{z}^\top \boldsymbol{z} \leqslant 1$ for $p \in (0, 1]$ and any $\boldsymbol{x} \in \Delta$. Equivalently, we require $(\boldsymbol{z}^\top \boldsymbol{z})^{\frac{1}{p+1}} \leqslant 1$ for $p \in (0, 1]$.

Note $(\boldsymbol{z}^\top \boldsymbol{z})^{\frac{1}{p+1}} = \left( \sum_i (U_i^{(p)} \boldsymbol{x})^{p+1} \right)^{\frac{1}{p+1}} = ||U^{(p)} \boldsymbol{x}||_{p+1}$. Therefore, we require

$$1 \geqslant \sup_{\boldsymbol{x} \in \Delta} ||U^{(p)} \boldsymbol{x}||_{p+1} \tag{10}$$

$$= \sup_{||\boldsymbol{x}||_1 = 1} ||U^{(p)} \boldsymbol{x}||_{p+1} \quad \text{for} \quad U^{(p)} \geqslant 0 \tag{11}$$

$$= ||U^{(p)}||_{1, p+1} \tag{12}$$

$$= \max_{j=1} ||U_{\cdot, j}^{(p)}||_{p+1} \quad \text{by Drakakis et al. (2009).} \tag{13}$$

**Remark.** *Among all admissible $U^{(p)}$, defining $U^{(p)}$ such that its columns have exactly unit $(p+1)$-norm achieves $\min_{U^{(p)}} \min_{\boldsymbol{x} \in \Delta} H_p^a(\boldsymbol{x})$.*

This follows from the previous remark and is desireable for the sake of defining a "tight" definition of entropy. Intuitively, by the conditions set thus far, $U^{(p)} = \boldsymbol{0}$ is admissible. Yet, this gives a loose definition of entropy where $H_a^p = 1/p$. It turns out that this intuition is required in the limit as $p \to 0$.

**Remark.** *$U^{(p)}$ must be precisely column stochastic for $H_a^p$ to remain finite in the limit of $p \to 0$.*

In the limit $p \to 0$, the denominator of $H_a^p$ goes to zero, therefore, by L'Hôpital's rule, the numerator must as well. The numerator goes to $z^\top z = \sum_i U_i^{(p)} x = \mathbf{1}^\top U^{(p)} x$. Therefore,

$$\forall x \in \Delta^{d-1} \quad 1 - \mathbf{1}^\top U^{(p)} x = 0. \tag{14}$$

Finite distributions only obey a single equality constraint, that is $x^\top \mathbf{1} = 1$, therefore it must be the case that $\mathbf{1}^\top U^{(p)} = \mathbf{1}^\top$, i.e., $U^{(p)}$ is column stochastic.

**Remark.** *$H_a^p$ is concave in $x$.*

Let $y_i = U_i^{(p)} x$. Then each element of the sum, $y_i^{p+1}$ is a convex function in $y_i$, which itself is a linear transformation on $x$. Therefore, $\sum_i (U_i^{(p)} x)^{p+1}$ is convex in $x$. Hence $H_a^p$ is concave in $x$.

**Remark.** *The gradients $\nabla_x H_a^p$ are well-defined.*

Recall (9), then:

$$\frac{\partial H_a^p}{\partial x_j} = -\frac{p+1}{p} \sum_i (U_i^{(p)} x)^p U_{ij}^{(p)} \tag{15}$$

$$\nabla_x H_a^p = -\frac{p+1}{p} (U^{(p)})^\top (U^{(p)} x)^p \tag{16}$$

which is well-defined for any choice of $U_{ij}^{(p)} \geqslant 0$ for all $i, j$.

**Remark.** *$H_a^p$ is well-defined in the limit as $p \to 0$, i.e., Shannon affinity entropy is well-defined.*

It is known that Shannon entropy can be recovered from Tsallis entropy in the limit as $p \to 0$. We repeat that derivation here and use L'Hôpital's rule. The derivative of the denominator is $1$, hence we find the limit is given by the finite derivative of the numerator:

$$\frac{d[pH_a^p]}{dp} = -\frac{d}{dp} \Big[ \sum_i y_i^{p+1} \Big] \tag{17}$$

$$= -\frac{d}{dp} \Big[ \sum_i e^{(p+1)\log(y_i)} \Big] \tag{18}$$

$$= -\sum_i \Big( \log(y_i) + (p+1)\frac{1}{y_i}\frac{dy_i}{dp} \Big) e^{(p+1)\log(y_i)}. \tag{19}$$

In the limit $p \to 0$, the derivative evaluates to

$$\frac{d[pH_a^p]}{dp} = -\sum_i \Big[ e^{(p+1)\log(y_i)} \log(y_i) \Big]\Big|_{p=0} - (p+1)\sum_i \Big[ \frac{1}{y_i}\frac{dy_i}{dp} e^{(p+1)\log(y_i)} \Big]\Big|_{p=0} \tag{20}$$

$$= -\sum_i y_i \log(y_i) - \sum_i \frac{dy_i}{dp}\Big|_{p=0} \tag{21}$$

$$= S(y) - \sum_i \frac{dy_i}{dp}\Big|_{p=0}. \tag{22}$$

**Remark.** *Let $K$ be a similarity matrix between actions with non-negative entries with positive column-sums. Then $U^{(p)} = K\,\mathtt{diag}\big(1/(\mathbf{1}^\top K^{p+1})^{1/(p+1)}\big)$ satisfies the conditions stated above for $U^{(p)}$.*

**Remark.** *Under the above choice of $U^{(p)}$, Shannon affinity entropy $S_a = H_a^{p \to 0}$ can be derived as:*

$$S_a(x) = S(U^{(0)} x) - \sum_j \Big[ \log(\sum_i K_{ij}) - \sum_i U_{ij}^{(0)} \log(K_{ij}) \Big] x_j. \tag{23}$$

The necessary $y_i$ term can be rewritten and its derivative (evaluated at $p = 0$) can be derived as follows:

$$y_i = U_i^{(p)} \boldsymbol{x} = \sum_j \frac{K_{ij}}{(\sum_{i'} K_{i'j}^{p+1})^{\frac{1}{p+1}}} x_j \tag{24}$$

$$= \sum_j K_{ij} x_j (\sum_{i'} K_{i'j}^{p+1})^{-\frac{1}{p+1}} \tag{25}$$

$$= \sum_j K_{ij} x_j e^{-\frac{1}{p+1} \log(\sum_{i'} K_{i'j}^{p+1})} \tag{26}$$

$$= \sum_j K_{ij} x_j e^{-\frac{1}{p+1} \log \left( \sum_{i'} e^{(p+1)\log(K_{i'j})} \right)} \tag{27}$$

$$\frac{dy_i}{dp} = \sum_j K_{ij} x_j e^{-\frac{1}{p+1} \log \left( \sum_{i'} e^{(p+1)\log(K_{i'j})} \right)} \left[ \frac{1}{(p+1)^2} \log \left( \sum_{i'} e^{(p+1)\log(K_{i'j})} \right) \right. \tag{28}$$

$$\left. - \frac{1}{p+1} \frac{1}{\sum_{i'} e^{(p+1)\log(K_{i'j})}} \sum_{i'} \log(K_{i'j}) e^{(p+1)\log(K_{i'j})} \right] \tag{29}$$

$$= \sum_j K_{ij} x_j (\sum_{i'} K_{i'j}^{p+1})^{-\frac{1}{p+1}} \left[ \frac{1}{(p+1)^2} \log(\sum_{i'} K_{i'j}^{p+1}) \right. \tag{30}$$

$$\left. - \frac{1}{p+1} \frac{1}{\sum_{i'} K_{i'j}^{p+1}} \sum_{i'} \log(K_{i'j}) K_{i'j}^{p+1} \right] \tag{31}$$

$$= \sum_j \left[ \frac{1}{(p+1)^2} \log(\sum_{i'} K_{i'j}^{p+1}) - \frac{1}{p+1} \sum_{i'} (U_{i'j}^{(p)})^{p+1} \log(K_{i'j}) \right] U_{ij}^{(p)} x_j \tag{32}$$

$$\frac{dy_i}{dp}\Big|_{p=0} = \sum_j \left[ \log(\sum_{i'} K_{i'j}) - \sum_{i'} U_{i'j}^{(0)} \log(K_{i'j}) \right] U_{ij}^{(0)} x_j \tag{33}$$

where we define $K_{ij} \log(K_{ij}) = 0$ if $K_{ij} = 0$ (which implies $(U_{ij}^{(p)})^{p+1} \log(K_{ij}) = 0$ if $K_{ij} = 0$).
Plugging this back into the second term in the formula for Shannon *affinity* entropy, we find

$$\sum_i \frac{dy_i}{dp}\Big|_{p=0} = \sum_i \sum_j \left[ \log(\sum_{i'} K_{i'j}) - \sum_{i'} U_{i'j}^{(0)} \log(K_{i'j}) \right] U_{ij}^{(0)} x_j \tag{34}$$

$$= \sum_j \left[ \log(\sum_{i'} K_{i'j}) - \sum_{i'} U_{i'j}^{(0)} \log(K_{i'j}) \right] x_j \sum_i U_{ij}^{(0)} \tag{35}$$

$$= \sum_j \left[ \log(\sum_{i'} K_{i'j}) - \sum_{i'} U_{i'j}^{(0)} \log(K_{i'j}) \right] x_j \tag{36}$$

completing the claim.

**Remark.** *In the case of duplicate strategies (clones), the maximizers of $H_a^p$ form precisely the set of distributions which arbitrarily distribute a mass of $\frac{1}{C}$ across each of the $C$ sets of clones.*

Consider the case of exact clones, i.e., $K$ is block diagonal (w.l.o.g.) with blocks of ones. Let there be $C$ clone groups each of size $n_c$ for $c \in \{1, \dots, C\}$. Let $c(i)$ map an action $i$ to its clone set. In this case, it can be shown that $U_{ij}^{(p)} = n_{c(i)}^{-\frac{1}{p+1}}$ if $c(i) = c(j)$, otherwise $U_{ij}^{(p)} = 0$. Note that the gradient of entropy w.r.t. $\boldsymbol{x}$ must be proportional to the ones vector for $\boldsymbol{x}$ to be a maximizer in the interior of the simplex. Let $\boldsymbol{x} = [\frac{1}{C}\boldsymbol{x}_1, \dots, \frac{1}{C}\boldsymbol{x}_C]$ with each $\boldsymbol{x}_c \in \mathbb{R}^{n_c}$ w.l.o.g. We will show that the set of maximizers of $H_a^p$ is necessarily the set of $\boldsymbol{x}$ where each $\boldsymbol{x}_c \in \Delta^{n_c-1}$. For $\boldsymbol{x}$ to be a maximizer, the gradient must be equal to the ones vector multiplied by a scalar $-d \in \mathbb{R}$:

$$\forall j \ \frac{\partial H_a^p(\boldsymbol{x})}{\partial x_j} = -\frac{p+1}{p} \sum_i (U_i^{(p)} \boldsymbol{x})^p U_{ij}^{(p)} \tag{37}$$

$$= -\frac{p+1}{p} \sum_i (\sum_k U_{ik}^{(p)} x_k)^p U_{ij}^{(p)} \tag{38}$$

$$= -\frac{p+1}{p} \sum_i \left(\frac{1}{C} n_{c(i)}^{-\frac{1}{p+1}} \mathbf{1}^\top \boldsymbol{x}_{c(i)}\right)^p U_{ij}^{(p)} \tag{39}$$

$$= -\frac{p+1}{p} n_{c(j)} \left(\frac{1}{C} n_{c(j)}^{-\frac{1}{p+1}} \mathbf{1}^\top \boldsymbol{x}_{c(j)}\right)^p n_{c(j)}^{-\frac{1}{p+1}} \tag{40}$$

$$= -\frac{p+1}{p} n_{c(j)} n_{c(j)}^{-\frac{p+1}{p+1}} \left(\frac{1}{C} \mathbf{1}^\top \boldsymbol{x}_{c(j)}\right)^p \tag{41}$$

$$= -\frac{p+1}{p} \left(\frac{1}{C} \mathbf{1}^\top \boldsymbol{x}_{c(j)}\right)^p = -d. \tag{42}$$

We also require $\boldsymbol{x} \in \Delta$, which implies

$$x_j \geqslant 0 \implies \boldsymbol{x}_{c(j)} \geqslant \mathbf{0} \tag{43}$$

$$1 = \sum_j x_j = \sum_c \frac{1}{C} \mathbf{1}_{n_c}^\top \boldsymbol{x}_c \tag{44}$$

$$= C \left(\frac{dp}{p+1}\right)^{1/p} = d^{1/p} C \left(\frac{p}{p+1}\right)^{1/p} \implies d = C^{-p} \left(\frac{p+1}{p}\right). \tag{45}$$

Finally, we know from (42)

$$\left(\frac{1}{C} \mathbf{1}^\top \boldsymbol{x}_{c(j)}\right)^p = \frac{dp}{p+1} = C^{-p} \tag{46}$$

$$\implies \mathbf{1}^\top \boldsymbol{x}_{c(j)} = 1 \tag{47}$$

proving the claim.

**Remark.** *In the case of duplicate strategies (clones), the maximizers of $H_a^p$ achieve an entropy value which is equal to the Tsallis entropy of the system with clones removed.*

If we evaluate the max entropy distribution we find

$$H_a^p(\boldsymbol{x}) = \frac{1}{p} \Big[ 1 - \sum_i (U_i^{(p)} \boldsymbol{x})^{p+1} \Big] \tag{48}$$

$$= \frac{1}{p} \Big[ 1 - \sum_i \left(\frac{1}{C} n_{c(i)}^{-\frac{1}{p+1}} \mathbf{1}^\top \boldsymbol{x}_{c(i)}\right)^{p+1} \Big] \tag{49}$$

$$= \frac{1}{p} \Big[ 1 - \sum_c n_c \left(\frac{1}{C} n_c^{-\frac{1}{p+1}} \mathbf{1}^\top \boldsymbol{x}_c\right)^{p+1} \Big] \tag{50}$$

$$= \frac{1}{p} \Big[ 1 - \sum_c n_c \left(\frac{1}{C} n_c^{-\frac{1}{p+1}}\right)^{p+1} \Big] \tag{51}$$

$$= \frac{1}{p} \Big[ 1 - \sum_c n_c n_c^{-1} \left(\frac{1}{C}\right)^{p+1} \Big] \tag{52}$$

$$= \frac{1}{p} \Big[ 1 - \sum_c \left(\frac{1}{C}\right)^{p+1} \Big] \tag{53}$$

which is precisely the Tsallis entropy of the uniform distribution over $C$ distinct clones.

## C   INTEGRALS OVER SIMPLEX

It is possible to derive a closed-form result for the dis-similarity kernel in (6) by appealing to known results of integrals of polynomial functions over the simplex.

Let $T^d = \{(x_1, \ldots, x_d) : x_i \geqslant 0, \sum_{i=1}^{d} x_i \leqslant 1\}$ be the standard simplex in $\mathbb{R}^d$. Let $\nu_i > 0$ for all $i$, then

$$\int_{T^d} x_1^{\nu_1 - 1} \ldots x_d^{\nu_d - 1} (1 - x_1 - \ldots - x_d)^{\nu_0 - 1} = \frac{\prod_{i=0}^{d} \Gamma(\nu_i)}{\Gamma(\sum_{i=0}^{d} \nu_i)}. \tag{54}$$

**Proposition C.1.** *From player $i$'s perspective, the expected dis-similarity between two actions $p$ and $q$ under a uniform distribution over all opponent joint strategy profiles $x_{-i}$ is equal to*

$$D_{pq}^{(i)} = \frac{1}{(d_i + 1)(d_i + 2)} \left[ ||U_p^{(i)} - U_q^{(i)}||^2 + \left(1^\top (U_p^{(i)} - U_q^{(i)})\right)^2 \right] \tag{55}$$

*where $U^{(i)}$ is a $|\mathcal{A}_i| \times |\mathcal{A}_{-i}|$ matrix where each entry $U_{a_i, a_{-i}}^{(i)}$ is the expected utility for player $i$ playing action $a_i$ against the background joint action $a_{-i}$. $U_{a_i}^{(i)}$ indicates an entire row of the matrix. The integer $d_i = \prod_{j \neq i} |\mathcal{A}_j|$.*

*Proof.* Recall (54) and $\Gamma(n) = (n-1)!$ for $n \in \mathbb{N}$. Let $r_p = \sum_w U_{pw} x_w$ be the rating for the $p$th action under an opponent strategy profile $x_{-i}$.

Then we want to compute $\mathbb{E}_{x_{-i} \sim Dir(\mathbf{1})}[(r_p - r_q)^2]$. Recall the volume of the simplex is $\frac{1}{d!}$. Then

$$\mathbb{E}_{x_{-i} \sim Dir(\mathbf{1})}[(r_p - r_q)^2] = \frac{\int_{T^d} (r_p - r_q)^2 dx_{-i}}{\int_{T^d} dx_{-i}} \tag{56}$$

$$= d! \int_{T^d} (r_p - r_q)^2 dx_{-i} \tag{57}$$

$$= d! \int_{T^d} (\sum_w U_{pw}^{(i)} x_w - \sum_w U_{qw}^{(i)} x_w)^2 dx_{-i} \tag{58}$$

$$= d! \int_{T^d} \left[ (\sum_w \sum_y U_{pw}^{(i)} U_{py}^{(i)} x_w x_y) + (\sum_w \sum_y U_{qw}^{(i)} U_{qy}^{(i)} x_w x_y) \right. \tag{59}$$

$$\left. - 2 (\sum_w \sum_y U_{pw}^{(i)} U_{qy}^{(i)} x_w x_y) \right] dx_{-i} \tag{60}$$

$$= d! \sum_w \sum_y \left[ \left( U_{pw}^{(i)} U_{py}^{(i)} \underbrace{\int_{T^d} x_w x_y dx_{-i}}_{\frac{2}{(d+2)!} \text{ if } w=y \text{ else } \frac{1}{(d+2)!}} \right) \right. \tag{61}$$

$$\left. + \left( U_{qw}^{(i)} U_{qy}^{(i)} \int_{T^d} x_w x_y dx_{-i} \right) - 2 \left( U_{pw}^{(i)} U_{qy}^{(i)} \int_{T^d} x_w x_y dx_{-i} \right) \right] \tag{62}$$

$$= \frac{d!}{(d+2)!} \sum_w \left[ (U_{pw}^{(i)2} + U_{qw}^{(i)2} - 2 U_{pw}^{(i)} U_{qw}^{(i)}) \right. \tag{63}$$

$$\left. + \sum_y \left( U_{pw}^{(i)} U_{py}^{(i)} + U_{qw}^{(i)} U_{qy}^{(i)} - 2 U_{pw}^{(i)} U_{qy}^{(i)} \right) \right] \tag{64}$$

$$= \frac{1}{(d+1)(d+2)} \left[ \sum_w (U_{pw}^{(i)} - U_{qw}^{(i)})^2 + (\sum_w U_{pw}^{(i)} - \sum_w U_{qw}^{(i)})^2 \right] \tag{65}$$

$$= \frac{1}{(d+1)(d+2)} \left[ ||U_p^{(i)} - U_q^{(i)}||^2 + \left(1^\top (U_p^{(i)} - U_q^{(i)})\right)^2 \right]. \tag{66}$$

$\square$

**Proposition C.2.** *From player $i$'s perspective, the expected dis-similarity between two actions $p$ and $q$ under a uniform distribution over all factorize-able opponent strategy profiles $x_{-i} = \prod_{j \neq i} x_j$ is equal to*

$$D_{pq}^{(i)} = \prod_{j \neq i} \frac{1}{(d_j + 1)(d_j + 2)} \Big( \tag{67}$$

$$\sum_{a_{-i} \in \mathcal{A}_{-i}} \sum_{a'_{-i} \in \mathcal{A}_{-i}} \left( u_i(p, a_{-i}) - u_i(q, a_{-i}) \right) \left( u_i(p, a'_{-i}) - u_i(q, a'_{-i}) \right) \left( 2^{\#a=a'} \right) \Big) \tag{68}$$

where the integer $d_i = |\mathcal{A}_i|$ and "$\#a{=}a'$" $= \sum_{j \neq i} \mathbf{1}[a_j = a'_j]$ indicates the number of action matches between two opponent profiles.

*Proof.* Let $r_p = \sum_{a_{-i} \in \mathcal{A}_{-i}} u_i(p, a_{-i}) \prod_{j \neq i} x_{j,a_j}$ be the rating for the $p$th action under an opponent profile $x_{-i} = \prod_{j \neq i} x_j$. Let $dx_{-i}$ be a shorthand for $dx_{-i}$. Likewise, let $\int_{T^{d_{-i}}}$ be a shorthand for $\int_{T^{d_1}} \cdots \int_{T^{d_{i-1}}} \int_{T^{d_{i+1}}} \cdots \int_{T^{d_n}}$.

Then we want to compute $\mathbb{E}_{x_j \sim Dir(\mathbf{1}) \forall j \neq i}[(r_p - r_q)^2]$. Recall the volume of a simplex is $\frac{1}{d!}$. Then

$$\mathbb{E}_{x_j \sim Dir(\mathbf{1}) \forall j \neq i}[(r_p - r_q)^2] \tag{69}$$

$$= \frac{\int_{T^{d_{-i}}} (r_i - r'_i)^2 dx_{-i}}{\int_{T^{d_{-i}}} dx_{-i}} \tag{70}$$

$$= \left(\prod_{j \neq i} d_j!\right) \int_{T^{d_{-i}}} (r_i - r'_i)^2 dx_{-i} \tag{71}$$

$$= \left(\prod_{j \neq i} d_j!\right) \int_{T^{d_{-i}}} \left(\sum_{a_{-i} \in \mathcal{A}_{-i}} u_i(p, a_{-i}) \prod_{j \neq i} x_{j,a_j} - \sum_{a_{-i} \in \mathcal{A}_{-i}} u_i(q, a_{-i}) \prod_{j \neq i} x_{j,a_j}\right)^2 dx_{-i} \tag{72}$$

$$= \left(\prod_{j \neq i} d_j!\right) \int_{T^{d_{-i}}} \left(\sum_{a_{-i} \in \mathcal{A}_{-i}} \prod_{j \neq i} x_{j,a_j} \big(u_i(p, a_{-i}) - u_i(q, a_{-i})\big)\right)^2 dx_{-i} \tag{73}$$

$$= \left(\prod_{j \neq i} d_j!\right) \int_{T^{d_{-i}}} \Big( \tag{74}$$

$$\sum_{a_{-i} \in \mathcal{A}_{-i}} \sum_{a'_{-i} \in \mathcal{A}_{-i}} \left(\prod_{j \neq i} x_{j,a_j}\right)\left(\prod_{j \neq i} x_{j,a'_j}\right) \big(u_i(p, a_{-i}) - u_i(q, a_{-i})\big)\big(u_i(p, a'_{-i}) - u_i(q, a'_{-i})\big)\Big) dx_{-i} \tag{75}$$

$$= \left(\prod_{j \neq i} d_j!\right) \int_{T^{d_{-i}}} \Big( \tag{76}$$

$$\sum_{a_{-i} \in \mathcal{A}_{-i}} \sum_{a'_{-i} \in \mathcal{A}_{-i}} \big(u_i(p, a_{-i}) - u_i(q, a_{-i})\big)\big(u_i(p, a'_{-i}) - u_i(q, a'_{-i})\big)\left(\prod_{j \neq i} x_{j,a_j} x_{j,a'_j}\right)\Big) dx_{-i} \tag{77}$$

$$= \left(\prod_{j \neq i} d_j!\right)\Big( \tag{78}$$

$$\sum_{a_{-i} \in \mathcal{A}_{-i}} \sum_{a'_{-i} \in \mathcal{A}_{-i}} \big(u_i(p, a_{-i}) - u_i(q, a_{-i})\big)\big(u_i(p, a'_{-i}) - u_i(q, a'_{-i})\big)\Big(\prod_{j \neq i} \underbrace{\int_{T^{d_j}} x_{j,a_j} x_{j,a'_j} dx_j}_{\frac{2}{(d_j+2)!} \text{ if } a_j = a'_j \text{ else } \frac{1}{(d_j+2)!}}\Big)\Big) \tag{79}$$

$$= \left(\prod_{j \neq i} d_j!\right)\Big/\left(\prod_{j \neq i} (d_j + 2)!\right)\Big( \tag{80}$$

$$\sum_{a_{-i} \in \mathcal{A}_{-i}} \sum_{a'_{-i} \in \mathcal{A}_{-i}} \big(u_i(p, a_{-i}) - u_i(q, a_{-i})\big)\big(u_i(p, a'_{-i}) - u_i(q, a'_{-i})\big)\big(2^{\#a=a'}\big)\Big) \tag{81}$$

$$= \prod_{j \neq i} \frac{1}{(d_j + 1)(d_j + 2)}\Big( \tag{82}$$

$$\sum_{a_{-i} \in \mathcal{A}_{-i}} \sum_{a'_{-i} \in \mathcal{A}_{-i}} \big(u_i(p, a_{-i}) - u_i(q, a_{-i})\big)\big(u_i(p, a'_{-i}) - u_i(q, a'_{-i})\big)\big(2^{\#a=a'}\big)\Big). \tag{83}$$

$$\square$$

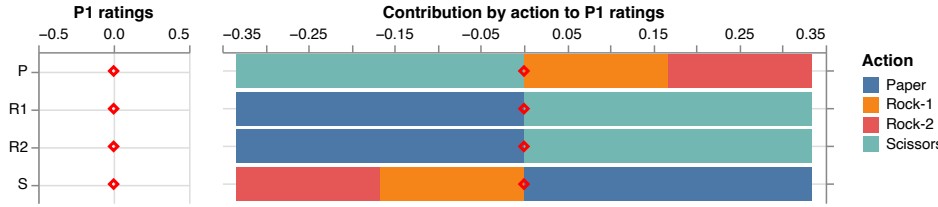

Figure 6: We visualise the marginal NE rating contributions of each player 2 action to each player 1 action. We show that a) all actions receive zero ratings and b) the rating of each action is interpretable and corresponds to our intuition.

# D    WARMUP: GAME-THEORETIC RANKING OF *rock-paper-scissors*

We provide a demonstration of game-theoretic ranking on the classic 2-player, 3-action zero-sum Rock-Paper-Scissors game. Balduzzi et al. (2018) proposed rating actions under the max-entropy Nash equilibrium of the game. In that case, each action receives a rating of zero. If we duplicate the Rock action, for example, the ratings remain zero under the max-entropy NE. Our proposed LLE based approach returns the same ratings.

|          | Rock    | Paper   | Scissors |
|----------|---------|---------|----------|
| Rock     | $0, 0$  | $-1, +1$ | $+1, -1$ |
| Paper    | $+1, -1$ | $0, 0$  | $-1, +1$ |
| Scissors | $-1, +1$ | $+1, -1$ | $0, 0$   |

|          | Rock1   | Rock2   | Paper   | Scissors |
|----------|---------|---------|---------|----------|
| Rock1    | $0, 0$  | $0, 0$  | $-1, +1$ | $+1, -1$ |
| Rock2    | $0, 0$  | $0, 0$  | $-1, +1$ | $+1, -1$ |
| Paper    | $+1, -1$ | $+1, -1$ | $0, 0$  | $-1, +1$ |
| Scissors | $-1, +1$ | $-1, +1$ | $+1, -1$ | $0, 0$   |

Figure 7: Rock-Paper-Scissors (RPS) Game and RPS Game with Duplicate Rock Action.

In Figure 6, we show that the equilibrium underlying the scalar ratings reflects incentive structure of the game — player 1 does not wish to deviate to the *Paper* action precisely because doing so would lead to losses against the *Scissors* action despite wins against the two *Rock* actions.

# E    VULNERABILITY OF STANDARD SHANNON ENTROPY

Prior work has shown max-entropy Nash equilibrium (equivalently max-entropy (C)CE) to be invariant to clones in 2-player zero-sum games (Balduzzi et al., 2018). We include a simple experiment here to illustrate why max-entropy Nash equilibrium becomes vulnerable to redundancy in the $N$-player general-sum setting.

**Chicken Game**    Consider the 2-player 2-action general-sum *Chicken* game. Let players receive $0$ if they both *swerve*. If one player swerves while the other goes straight, the one who swerves receives $-1$ and the other $+1$. If both go straight, then they both receive $-12$. This game has three NEs. Two are pure in which one player goes straight and the other swerves. The third is symmetric and the max-entropy NE of this game; each player swerves with probability $^{11}/_{12}$. Both *straight* and *swerve* have an expected payoff of $-1/12$ under this NE. If we duplicate the *straight* action, the original max-entropy NE becomes the *min*-entropy NE! The other two NEs representing each player swerving while the other goes straight now have higher entropy. The player that swerves rates their swerve and straight actions as $-1$ and $-12$ respectively. The player that goes straight rates their swerve and straight actions as $0$ and $1$ respectively, demonstrating that the max-entropy NE solution concept is not invariant to clones in the general-sum setting.

The story in the max-entropy CCE setting is more nuanced. We find that although the CCE ratings change under the addition of clones, the ratio of the ratings of the two actions remains stable. Further investigation is necessary to understand whether max-entropy CCE ratings are equivariant (robust up to affine transformations of the ratings) to cloned actions.

By contrast, we show in Figure 9 that all actions would receive zero ratings under our proposed equilibrium ratings. In other words, our equilibrium selection procedure continues to select the

|         | Swerve | Straight   |
|---------|--------|------------|
| Swerve  | $0, 0$ | $-1, +1$   |
| Straight| $+1, -1$ | $-12, -12$ |

|          | Swerve   | Straight    | Straight    |
|----------|----------|-------------|-------------|
| Swerve   | $0, 0$   | $-1, +1$    | $-1, +1$    |
| Straight | $+1, -1$ | $-12, -12$  | $-12, -12$  |
| Straight | $+1, -1$ | $-12, -12$  | $-12, -12$  |

Figure 8: Chicken Game and Chicken Game with Duplicate Straight Actions.

Figure 9: We visualise the marginal NE rating contributions of each player 2 action to each player 1 action. We show that a) all actions receive zero ratings and b) the rating of each action is interpretable and corresponds to our intuition.

mixed-strategy NE in the original game, unaffected by the additional redundant "straight" action. Further, the widths of the bars are interpretable: suggesting that deviating to the *Swerve* action is a safe option without major risk or reward. Deviating to one of the *Straight* actions however, can lead to high rewards but also catastrophic losses.

# F  EXPERIMENTS

## F.1  SIMULATED MODEL AND PROMPT IMPROVEMENT PATH

Algorithm 1 describes our simulated model and prompt improvement procedure. At each iteration, we add a new prompt and a model following an evolutionary procedure. We require all prompts to be probability distributions over skill dimensions. We model for a transitive dimension for models by representing each model vector as a sum of probability vectors over skills. A new model is added to the set of models $\mathcal{A}_m$ if and only if it becomes top-ranked according to the rating function $r$. A new prompt is added as long as it is the best-of-$P'$ sampled prompts and does not have to be top-ranked.

## F.2  EQUILIBRIUM-SOLVING HYPER-PARAMETERS

We use the same set of hyper-parameters for all our experiments. For affinity-entropy $H_a^p(x)$, we use $p = 1$ and set kernel variance to $1\mathrm{e}{-6}$. To solve for a max affinity-entropy distribution we use gradient descent. The max affinity-entropy distribution is then used in NE and CCE solving.

For NE solving using LLE approximation, we initialize temperature $\tau = 1.0$ which is annealed exponentially with a decay rate of $0.95$ every $250$ gradient updates if and only if the exploitability in the annealed game $\mathcal{L}^\tau(x)$ (Equation (4)) is at most $1\mathrm{e}{-5}$. We set the terminal temperature to $\tau = 1\mathrm{e}{-2}$. We early terminate the equilibrium solving if we have found an $\epsilon-$NE with $\epsilon = 1\mathrm{e}{-3}$. For CCE solving, the optimization problem is convex and we minimize Equation 8 directly. For gradient descent, we use an Adam optimizer Kingma (2014) with a fixed learning rate $1\mathrm{e}{-2}$ for all steps (maximizing affinity-entropy and equilibrium solving).

## F.3  THE `arena-hard-v0.1` EVALUATION DATA

We evaluate our method on the `arena-hard-v0.1` dataset (Li et al., 2024b) with 500 prompts and 17 competing models. The set of prompts as well as model responses are downloaded from LMSYS data repository (https://huggingface.co/spaces/lmsys/arena-hard-browser), with the exception of `gemini-1.5-pro-api-0514` and `gemini-1.5-flash-api-0514`. As we need to tabulate the payoff tensor for all model pairs, we sampled 8 preference ratings using `gemini-1.5-pro-api-0514` for each model pair, with 4 samples for each permutation to account for potential position bias of the LLM rater. Pairwise model utility is averaged over all ratings samples.

---

**Algorithm 1** Evolutionary model and prompt selection procedure

---

1: Let $K$ be the number of orthogonal skill dimensions.
2: Let $r : \mathcal{A}_p \times \mathcal{A}_m \to \boldsymbol{r}_p, \boldsymbol{r}_m$ be a rating function assigning a scalar rating to each action.
3: Let $P_0$, $M_0$ be the number of initial prompts and models.
4: Let $P'$, $M'$ be the number of sampled candidate prompts and models at each iteration.
5:
6: $\mathcal{A}_p^0 \sim \mathrm{Dirichlet}(\mathbf{1}_{1:K}, P_0)$           $\triangleright$ $P_0$ sampled initial prompts.
7: $\mathcal{A}_m^0 \sim \mathrm{Dirichlet}(\mathbf{1}_{1:K}, M_0)$           $\triangleright$ $M_0$ sampled initial models.
8:
9: **for** $t \in [1, \dots]$ **do**
10:     **if** additional prompts **then**           $\triangleright$ If adding new prompts.
11:         $\mathcal{A}_p' \sim \mathrm{Dirichlet}(\mathbf{1}_{1:K}, P')$           $\triangleright$ Sampling $P'$ candidate prompts.
12:         $\boldsymbol{r}_{p, \_} \leftarrow r(A_p' \cup A_p, A_m)$
13:         $\mathcal{A}_p \leftarrow \mathcal{A}_p \cup \{A_p'[\arg\max \boldsymbol{r}_p[: P']]\}$      $\triangleright$ Add best-of-$P'$ prompt.
14:     **end if**
15:     $\mathcal{A}_m' \leftarrow \mathbf{0}$
16:     **while** true **do**
17:         $\Delta_m \leftarrow \mathrm{Dirichlet}(\mathbf{1}_{1:K}, M')$     $\triangleright$ Sampling $M'$ model improvement vectors.
18:         $\_, \boldsymbol{r}_m \leftarrow r(A_p, \{\mathcal{A}_m' + \Delta_m\} \cup A_m)$     $\triangleright$ Evaluate improved candidate models.
19:         $\mathcal{A}_m' \leftarrow \mathcal{A}_m' + \Delta_m[\arg\max \boldsymbol{r}_m[: M']]$
20:         **if** $\arg\max \boldsymbol{r}_m[: M'] = \arg\max \boldsymbol{r}_m$ **then**
21:             $\mathcal{A}_m = \{\mathcal{A}_m'[\arg\max \boldsymbol{r}_m]\} \cup \mathcal{A}_m$     $\triangleright$ Add a new top-ranked model.
22:             **break**
23:         **end if**
24:     **end while**
25: **end for**

---

Table 1: Prompt and king actions that each define 16 pure-strategy Nash equilibria — any rebel action except the model played by the king player is a pure-strategy NE.

| Prompt | King |
|---|---|
| "Can you implement a python tool that is intended to ru..." | gemini-1.5-pro-api-0514 |
| "Hi. I have this URL which I can paste in my Microsoft ..." | gemini-1.5-pro-api-0514 |
| "Please provide a simple RESPONSE to the following PROM..." | claude-3-5-sonnet-20240620 |
| "Take on the rol eof an Gherkin expert. Can you improve..." | claude-3-5-sonnet-20240620 |
| "Write a small python function that get all the links o..." | gemini-1.5-flash-api-0514 |

F.4    RISK-DOMINANT EQUILIBRIA

Our `king-of-the-hill` evaluation game admits a multitude of Nash equilibria, among them 80 are pure-strategy NEs (see Table 1). Additionally, we computed 128 mixed-strategy NEs with exploitability at most $1\mathrm{e}{-}2$ that each derives a distinct set of ratings. In particular, one of the 128 mixed-strategy NEs is pre-computed by our NE solving and selection procedure by tracing the QRE continuum, which we refer to as the 0-th equilibrium, or $\boldsymbol{x}^0$.

A longstanding challenge in game theory is that of equilibrium selection. Suppose that every player knows that there are many equilibria in the game, each player must confront the following question during play: out of all equilibria, which equilibrium strategy should I play and relatedly, which equilibrium would each of my co-players play? This is critical, as miscoordinating could lead to arbitrarily bad outcome, despite each player playing one of its equilibrium strategies. For instance, everyone driving on the right or left hand side of the road are two valid equilibria, but miscoordinating would be devastating.

It is for this reason that the notion of risk-dominance of Harsanyi & Selten (1988) is critically important: the Nobel-prize winning theorem suggests that players would each iterate on their prior beliefs over which equilibria its co-players would play and choose the one that is the least *risky* when players *miscoordinate* under such priors. Here, we show empirically that our solution concept leads to risk-dominant equilibria as suggested by Herings & Peeters (2010). To do so, we simultaneously minimize the exploitability of several profiles in parallel with a regularizer that maximizes the $L_2$

rating differences between any two profiles by gradient descent as in Liu et al. (2024). This yields an additional 127 NEs with exploitability at most $1e{-}2$ that we analyze in Figure 10.

Figure 10 *(Top)* shows the 128 mixed-strategy NEs with distinct model ratings. Figure 10 *(Center)* shows the expected payoffs to player $i$ when it plays its $p$-th equilibrium strategy $x_i^p$ when other players uniformly choose one of theirs, or $\mathbb{E}_{q \sim \pi_u} \left[ u_i(x_i^p, x_{-i}^q) \right]$ with $\pi_u$ a uniform distribution over 128 equilibria. In yellow, we show the sum of per-player expected payoffs. We confirm that many NEs are indeed *risky*, as their stability relies heavily on all players coordinating on the same equilibrium. Figure 10 *(Bottom)* takes things one step further and follows the intuition of risk dominance more closely. Starting from a uniform prior belief over player $i$'s choice of equilibria, $\pi_i^0 = \pi_u$, each player iterate their believes over other players' choices of equilibrium based on the expected payoff of them playing each equilibrium.

Specifically, we let

$$\pi_i^{t+1} = \text{softmax}\left( \log \pi_i^t + \eta \mathbb{E}_{\substack{\forall j \neq i \\ t(j) \sim \pi_j}} \left[ \boldsymbol{u}_i(\ldots, x_{i-1}^{t(i-1)}, x_{i+1}^{t(i+1)}, \ldots) \right] \right) \tag{84}$$

with $\eta = 1e{-}2$ the step-size and we compute the expected payoffs to player $i$ when playing its $k$-th equilibrium at $T = 10,000$ as

$$\mathbb{E}_{\substack{\forall j \neq i \\ e(j) \sim \pi_j^T}} \left[ u_i(\ldots, x_{i-1}^{e(i-1)}, x_i^k, x_{i+1}^{e(i+1)}, \ldots) \right] \tag{85}$$

Ordered by the sum of expected payoffs for all players, we observe that the Nash equilibrium our procedure selects (equilibrium $\boldsymbol{x}^0$) is the least risky among 128 mixed-strategy NEs of the game, without any player being particularly worse off than others even when players miscoordinate.

## F.5 INVARIANT EVALUATION

We show in Figure 11 the effect of introducing *near* redundant adversarial prompts on the equilibrium ratings. While our invariant property is limited to exact clones, our results show that our approach results in rankings that degrade gracefully in this approximate case, even with 1,000 adversarial prompts. The Elo rating system suffers from such bias in data similarly as in the exact case Figure 3.

In Figure 12 we provide a detailed breakdown of our NE and CCE ratings results (without redundant adversarial prompts). We show the actions of each player ranked by their equilibrium ratings and by their support under the equilibrium marginal distribution.

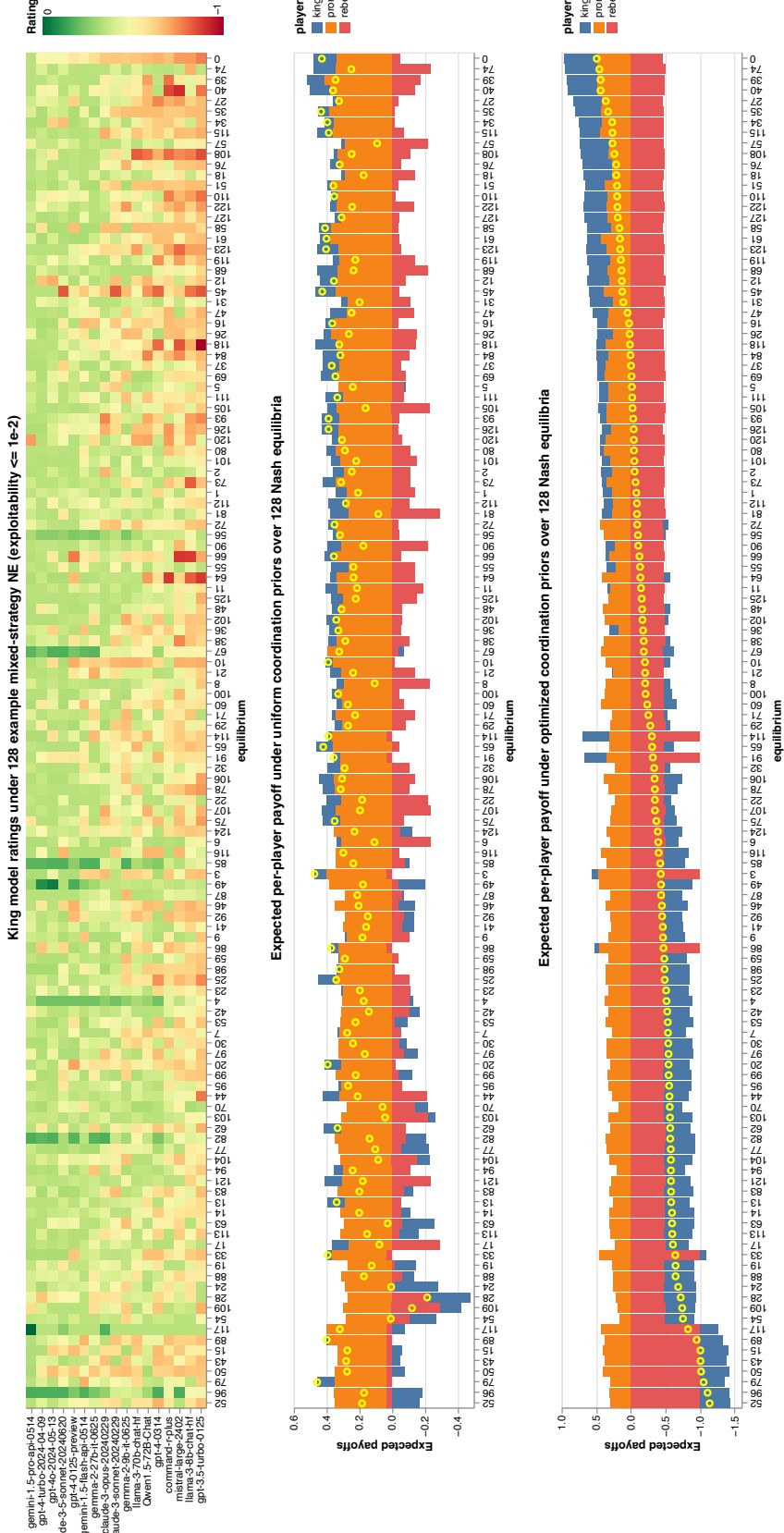

Figure 10: From top to bottom: a) we show the distinct king player action ratings derived from 128 mixed-strategy NEs of the king-of-the-hill game. All NEs have exploitability at most $\epsilon \leqslant 1e-2$; b) we show the expected payoff to each player under uniform priors over their 128 equilibria; yellow circles show the sum of expected per-player payoffs; c) we show the same analysis as in b) but the expectation is taken under optimized equilibrium priors. Equilibrium 0 (rightmost) is the LLE our NE solving procedure select.

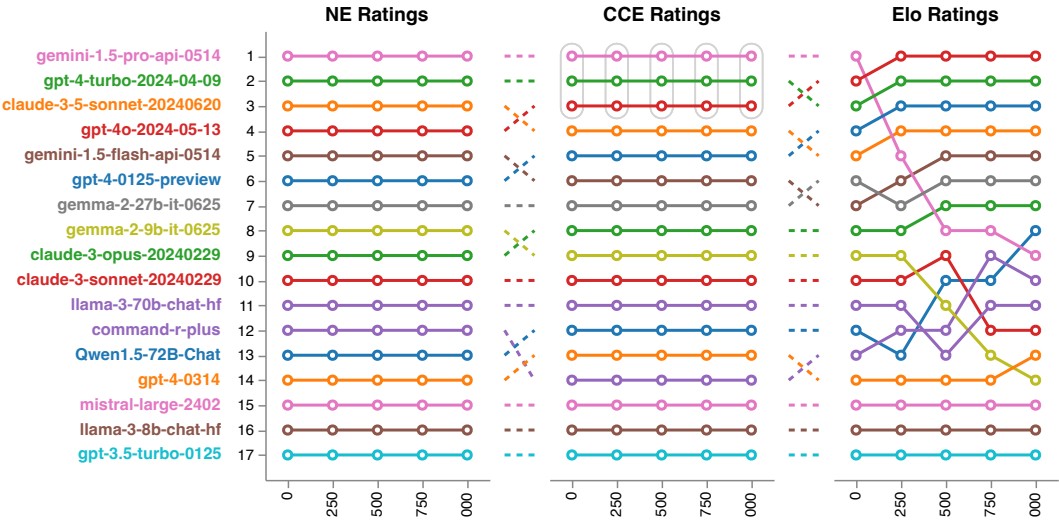

Figure 11: We introduce an increasing number of redundant copies of prompts adversarial to `gemini-1.5-pro-api-0514` with noise sampled from Uniform$(-0.01, 0.01)$ applied to their payoffs. Equilibrium ratings with a clone invariant selection procedure degrades gracefully to noisy redundancy while the Elo ratings become incrementally skewed. Models at the same rank (with an absolute rating difference at most $1e-4$) are grouped in grey and ordered alphabetically. We caveat that the specific rankings reported are subject to the LLM preference model used which in this case may exhibit a self-preference to the Gemini family of models.

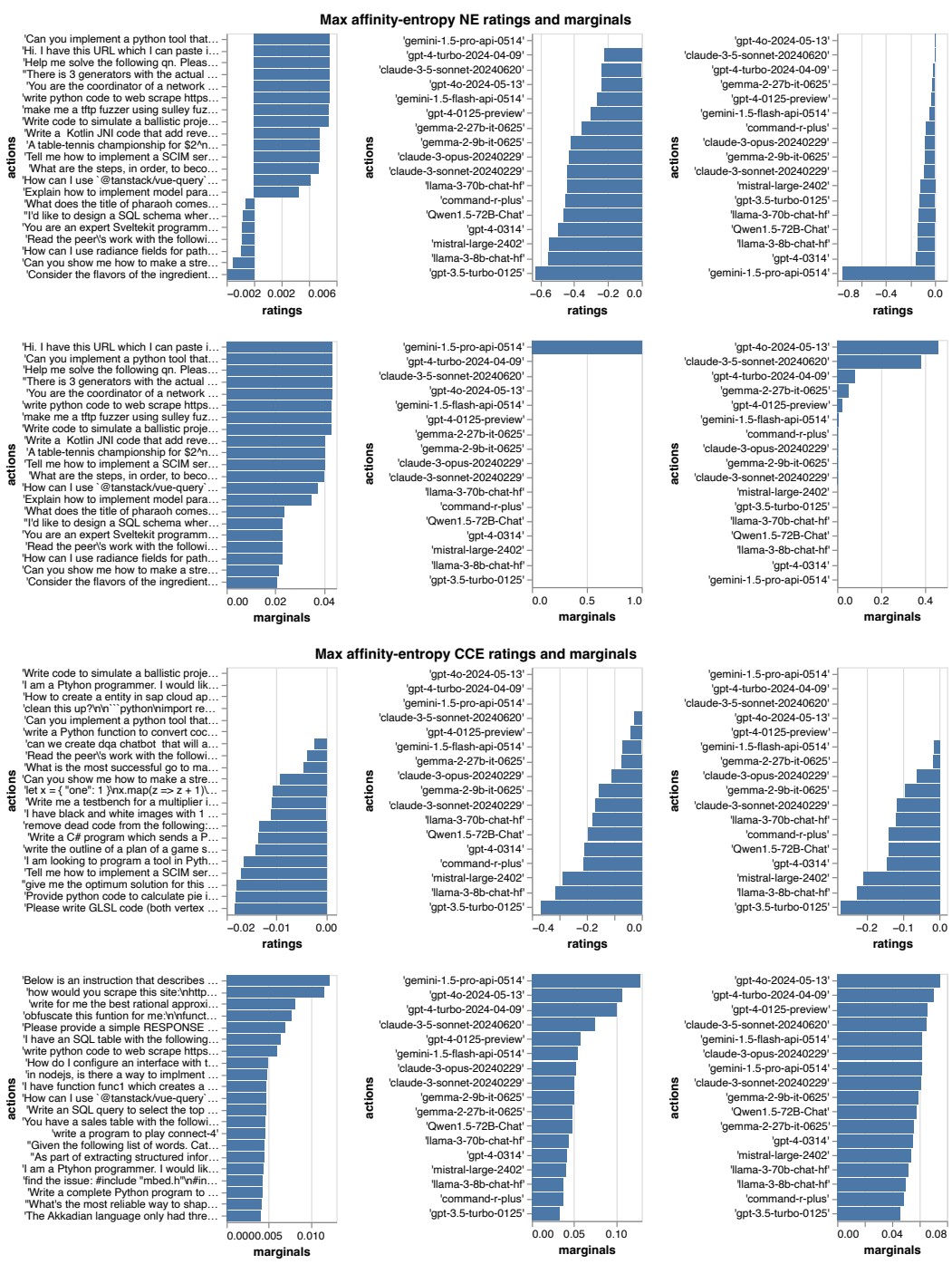

Figure 12: We show actions of each player ranked by their rating and equilibrium support under NE *(Top)* and CCE *(Bottom)* profiles respectively.

