# OpenReview forum: "Re-evaluating Open-ended Evaluation of Large Language Models"
_ICLR.cc/2025/Conference — ICLR 2025 Poster_

### Official Review · Reviewer_56qC · 2024-10-28

**Soundness:** 3
**Presentation:** 2
**Contribution:** 3
**Rating:** 6
**Confidence:** 3

**Summary:**

The current evaluation focuses on assessing a specific ability, but the emergence of technologies like LLMs has made the evaluation of open-ended tasks more important. Existing methods use Elo ratings for evaluation, but this approach's sensitivity to redundancy may lead to biased assessments. The authors propose a game theory-based evaluation method to address this issue. Experimental results indicate that this method can effectively identify prompts that can be distinctly differentiated and is less sensitive to redundancy compared to Elo-based methods.

**Strengths:**

The motivation of this paper is relatively meaningful, and studying the impact of evaluation on the development of LLMs is indeed an important issue, especially for open-ended evaluation.

Using game theoretical methods to assess the open-ended response capabilities of LLMs seems to be a relatively novel approach.

**Weaknesses:**

The article severely lacks background information, leaving readers quite confused about its position in the current research landscape. The Background section is too general; at least, it should introduce what 'king-of-the-hill' used in the actual modeling of this paper is. The Related Works section also fails to fulfill its intended purpose and function in the whole paper. I believe there are far more rating methods than those mentioned (such as [1][2] mentioned later). The differences between the method in this paper and other rating methods, other Elo-based methods, and other game-theoretic methods are not explained in detail. Moreover, the contribution of this paper is unclear. What specific shortcomings of the Elo-based method does it aim to address? Is it the robustness and sensitivity to redundancy mentioned in the abstract, or the poor skill assessment mentioned in section 1.1? Given the unclear statements in this paper, it is difficult for me to evaluate its contribution. If there are any misunderstandings on my part, I welcome further discussion at a later stage.

The paper emphasizes that this method can be applied to fully open skill assessments, but I believe its openness and generality come from the scoring model based on LLM (such as the gemini-1.5-pro mentioned in the paper). This is not a satisfactory solution, and what are the essential advantages of this over traditional LLM-eval-based methods (such as AlpacaEval [1][2])?

Figure 3 is confusing and lacks explanations for the connections. Figure 5 is even more perplexing, as it is unclear what the purpose of displaying these color blocks is and what it is trying to convey.

The experimental section lacks a comparative baseline. How much advantage does this evaluation method have over existing methods (such as various Elo-based methods and previous game-theoretic methods mentioned in the related works section)? Additionally, the dataset used in the experiments of this paper consists of only 500 questions, which is not a sufficiently convincing amount of data.


**Reference**

[1] AlpacaFarm: A Simulation Framework for Methods that Learn from Human Feedback (https://arxiv.org/abs/2305.14387)

[2] Length-Controlled AlpacaEval: A Simple Way to Debias Automatic Evaluators (https://arxiv.org/abs/2404.04475)

**Questions:**

Around line 60, why is this setting a necessity? I understand that using game-theoretic methods might have certain advantages, but why must this paper choose this type of method?

This paper uses gemini-1.5-pro as the judger. What considerations went into this choice? Why not use other more powerful models?

The text repeatedly mentions skill, skill entropy. What exactly is the definition of skill in this paper? Is it capability? If so, why can't we use targeted evaluations, or LLM-eval (in fact, I think this paper is also a type of LLM-eval method), and must instead use Elo-based methods?

For other questions see the **Weaknesses** section.

---

> ### Author Response · Authors · 2024-11-18
>
> We thank reviewer 56qC for recognising the relevance and urgency of our work. We also thank the reviewer for highlighting several points of confusion which has helped us improve our manuscript.
>
> We would like to start by clarifying what we meant by *open-ended evaluation*. We believe the reviewer understood the notion of “open-ended evaluation” in the sense that LLMs exhibit *open-ended* language capability. This is not our intention. In fact, our method applies more generally, for example, to the setting of RL agent evaluation by Atari games, as in [1]. Instead, *open-ended evaluation* refers to evaluation systems such as LMSYS where the set of prompts (equiv., environments/tasks) or models (equiv., agents) come from **open-ended** user submission: anyone can freely submit evaluation prompts with minimal gatekeeping from the LMSYS designers. Note, this is not how evaluation systems have worked traditionally; MNIST, ImageNet, Kaggle, Atari environments, etc. are all carefully curated and controlled evaluation benchmark sets. This new *open-ended evaluation* setting requires new thinking and tools.
>
> As such, we argue that if anyone can submit a large number of redundant or near-redundant prompts, the rating methodology ought to be robust against intentional **or** accidental redundancy. We then demonstrate in Figure 3 that the Elo rating system does not have this property and we can manipulate the rating of the Gemini model arbitrarily.
>
> It is also in this sense that game-theory is a natural direction to investigate: having 5 copies of rock in a rock-paper-scissors game does not change the fact that at an equilibrium, each action is still played ⅓ of the time and all actions would receive an equilibrium rating of zero whereas the paper action would look like a much better choice if Elo ratings are used.
>
> **> Q1: The Background section is too general; … The Related Works section also fails to fulfill its intended purpose and function in the whole paper. …**
>
> Although we have motivated our work in the context of LLM evaluation, with LMSYS one of the most prominent open-ended evaluation systems, we believe our work should be compared to prior works that propose rating methodologies rather than their applications in a specific domain --- our method is not restricted to LLM evaluation nor does it require LLMs in its use. For instance, in a rock-paper-scissors game, our method equally applies, as we have done so in Appendix D to provide intuition. It is for this reason that the description of the 'king-of-the-hill' game is relegated to the method description, as it’s only an example of gamification that can and should be reconsidered for different application domains.
>
> Nevertheless, we agree with the reviewer that the Alpaca papers should be cited, given their prominence in the broader LLM evaluation landscape and we have cited them in our latest revision.

---

> ### Author Response · Authors · 2024-11-18
>
> **> Q2: The differences between the method in this paper and other rating methods, other Elo-based methods, and other game-theoretic methods are not explained in detail.**
>
> In the Related Works section we have made references to Elo-based rating methods, methods based on Social Choice Theory and methods based on game-theoretic solution concepts. The differences between our approach and Elo-based methods are discussed throughout our experiments. The differences between our work and [1] has been discussed in detail in L50-L60.
>
> Nevertheless, we recognize that we can be more specific in our discussions in the Related Works section and we have revised accordingly in our latest revision.
>
> **> Q3: What specific shortcomings of the Elo-based method does it aim to address? Is it the robustness and sensitivity to redundancy mentioned in the abstract?**
>
> Yes.
>
> Section 1.1 (L135-143) suggests a non-obvious failure mode of hill-climbing on Elo scores **due to their sensitivity to redundancy**. If the set of openly crowd-sourced prompts over-represent one capability over another, then we might not see well-rounded improvement of models since specializing on the over-represented skill can be more effective.
>
> **> Q4: what are the essential advantages of this over traditional LLM-eval-based methods (such as AlpacaEval)**
>
> We hope our clarification and revision in the manuscript would help address this point of confusion. AlpacaEval family of works are *not*, according to our interpretation, open-ended evaluation. Specifically, the set of evaluation prompts are curated by the authors and we trust the authors would apply reasonable precautions to ensure a certain level of quality of the evaluation prompt set (e.g. removal of exact or near redundant prompts). In this case, ranking by *average* win-rate can be effective.
>
> Our method offers additional benefits beyond redundancy too:
> 1. it provides quality ratings of prompts (see Figure 4)  and;
> 2. it provides interpretable ratings (see Figure 5 and Appendix D-E which shows how one player’s actions contribute to another player’s action ratings).
>
> Indeed, these benefits could be helpful in the AlpacaEval setting too to highlight discriminative prompts and to understand the comparative strengths and weaknesses of each model.
>
>
> **> Q5: Figure 3 is confusing…**
>
> Figure 3 shows that one can intentionally and arbitrarily manipulate the Elo rating of a model (in this case `gemini-1.5-pro`) by introducing replicas of prompts where `gemini-1.5-pro` performed poorly. Note that for LMSYS evaluation, this type of attack is in principle feasible by a motivated actor. Our equilibrium ratings are invariant to such manipulation. NE(-a) [1] and CCE(-a) [2] show the necessity of the affinity entropy we introduced.
>
> Reading from the top of Figure 3, it suggests that `gemini-1.5-pro` remains top-ranked under NE ratings and CCE ratings. The dotted line shows it is also top-ranked according to Elo ratings. However, its rating decreases arbitrarily, as we duplicate prompts where `gemini-1.5-pro` performed poorly.
>
> We hope this helps and we have revised the Figure caption for clarity.

---

> ### Author Response · Authors · 2024-11-18
>
> **> Q6: Figure 5 is confusing…**
>
>
> We have now limited the visualisation to show the top-5 models only and we agree the earlier version is overwhelming and significantly simplified Figure 5 by grouping rebel models into family of models which revealed interesting game dynamics. We have also performed the same visualisation in Appendix D in a game of rock-rock-paper-scissors to help provide intuition.
>
> Intuitively, each king model receives a scalar equilibrium rating, which can be explained by and is a sum of contributions by each rebel model. The rebel model making the most negative (positive) contribution is the one that the king model performs the worst (best) against relative to its equilibrium strategy.
>
> **> Q7: How much advantage does this evaluation method have over existing methods (such as various Elo-based methods and previous game-theoretic methods mentioned in the related works section)?**
>
> Our proposed methods are invariant to redundancy; prior Elo-based methods and game-theoretic methods are not. We do not have a quantitative measure of quality for rating methods. However, we believe a good rating method should be one that is suitable for the community to hillclimb on.
>
>
> We argue that Elo-based ratings are not:
>
> 1) it can be manipulated by introducing redundant prompts (Figure 3) and;
> 2) it does not lead to well-rounded improvement (Figure 1). Figure 3 (Right) also show that naive equilibrium ratings are not clone-invariant [1, 2].
>
> We argue that our equilibrium rating methods, combined with our definition of affinity entropy, provides ratings that are robust to redundancy and as a result, lead to well-rounded improvements on diverse skills.
>
> **> Q8: The dataset is too small at 500 prompts.**
>
>
> To the best of our knowledge, ArenaHard 500 is the largest publicly available prompt set for which we have model responses from a diverse set of 17 models for each prompt. This lets us pairwise compare all models on all prompts using an automated judge LLM.
>
>
> We believe the size of the dataset is sufficient to make the point that our method is clone-invariant and provides intuitive rankings of models. However, we agree with the reviewer that 500 is too small to suggest that our method can scale. If the reviewer is concerned about our method being able to scale to larger prompt sets, we can include a runtime analysis for a much larger (synthetic) dataset in a follow-up revision.
>
> Nevertheless, please let us know if there are other datasets we should be aware of.

---

> ### Author Response · Authors · 2024-11-18
>
> **> Q9: Around line 60, why is this setting a necessity?**
>
>
> We believe this is a misunderstanding. On L59 we italicised “adversarial” and the “necessity” refers to the need for Balduzzi 2018 [1] to restrict themselves to a two-player zero-sum gamification. We did not argue that game-theoretic evaluation is a necessity but merely that Balduzzi 2018 modelled evaluation as a two-player zero-sum agent-vs-task game out of necessity as max-entropy Nash equilibrium is no longer unique beyond this setting.
>
>
> We thank the reviewer for pointing out this source of confusion and we have revised our writing in the latest revision.
>
>
> **> Q10: This paper uses gemini-1.5-pro as the judger. Why not more powerful models?**
>
>
> The main reason is that the Gemini Pro model is relatively affordable and reasonably capable at the time of our experiment. We are aware that alternative models exist and that LLM judges exhibit self-preferences and we have caveated accordingly in L200-L203.
>
>
> **> Q11: The text repeatedly mentions skill, skill entropy. What exactly is the definition of skill in this paper?**
>
>
> We believe discussions around the notion of “skill entropy” are primarily restricted to Section 1.1 where we proposed a simulated thought experiment that serves to demonstrate the potential danger of chasing after ever-higher Elo ratings. Please see L88-L90 for the setting of this simulated experiment for a definition of a skill.
>
>
> In real-world LLM development, model developers may optimise models for other considerations than improving on LMSYS Elo ratings and users who submit prompts may not solely focus on *separability*. As such, we believe a simulated thought experiment can provide insights more clearly and help draw attention towards more principled interpretation of ever-increasing Elo scores.
>
>
> To summarize, Section 1.1 shows that equilibrium ratings promote well-rounded model improvements across skills whereas models can always improve their Elo ratings by improving on a single skill dimension that is over-represented in the prompt set.
>
> ---
>
> [1] David Balduzzi, Karl Tuyls, Julien Perolat, and Thore Graepel. Re-evaluating evaluation. Advances in Neural Information Processing Systems, 31, 2018.
>
> [2] Luke Marris, Marc Lanctot, Ian Gemp, Shayegan Omidshafiei, Stephen McAleer, Jerome Connor,
> Karl Tuyls, and Thore Graepel. Game theoretic rating in n-player general-sum games with equilibria, 2022b. URL https://arxiv.org/abs/2210.02205.
>
> [3] Tianle Li, Wei-Lin Chiang, Evan Frick, Lisa Dunlap, Tianhao Wu, Banghua Zhu, Joseph E. Gonzalez, and Ion Stoica. From crowdsourced data to high-quality benchmarks: Arena-hard and
> benchbuilder pipeline, 2024a.

---

> > ### Comment · Reviewer_56qC · 2024-11-20
> >
> > Thank you for your detailed explanation. Based on your clarification, I have reviewed your revised version again and now have a clearer understanding of the methodology and contributions of this paper. I now believe this is a good piece of work, and I have increased the rating.

---

### Official Review · Reviewer_itPT · 2024-11-04

**Soundness:** 3
**Presentation:** 2
**Contribution:** 3
**Rating:** 6
**Confidence:** 4

**Summary:**

This paper explores a new approach to evaluating LLMs in open systems to address the bias introduced by Elo, proposing an evaluation framework based on game theory that views the evaluation process as a three-player game involving the " promptplayer," "king player," and "rebelplayer." By introducing novel solutions applicable to both general and game settings, the proposed method is more robust in handling data redundancy and excels in balancing model skills. Experimental results demonstrate the effectiveness of this approach.

**Strengths:**

This paper is good. Firstly, from a technical perspective, it introduces a 3-player game model based on game theory, providing equilibrium solutions applicable to N-player games. This is highly innovative in my opinion. Moreover, the paper conducts an analysis based on this paradigm, utilizing Affinity Entropy to meet assumptions and thereby addressing the limitations of traditional Elo evaluation methods in handling redundant data and biases.

Secondly, there is a very close integration between the main text and the experimental section of the paper. The theoretical design of the game theory model and its experimental validation complement each other, showcasing the effectiveness of the method on real-world datasets. The authors demonstrate the advantages of the Affinity Entropy method through theoretical analysis, which is further supported by experimental data, ensuring a coherent and consistent research logic.

Lastly, in terms of experimental design, the authors conducted thorough validation on real open LLM evaluation datasets, demonstrating the robustness and wide applicability of the method. The introduction of various comparison methods and key evaluation metrics adds to the robustness of the study.

**Weaknesses:**

There are a few points that confuse me:
1. In the background section, the introduction to NE and CCE seems somewhat limited, although this is a focal point of the paper. Insufficient explanation might lead to misunderstandings upon re-reading. LLE is also an important concept, yet it is only mentioned in section 3.2, causing some logical confusion.
2. In the paper, what specific meaning does "p" represent in the context of Affinity Entropy? Is it reasonable to directly set p=1 in Theorem 1? The subsequent appearance of p in D_{pq} further adds to the confusion.
3. Could it be beneficial to consider including more benchmark comparison methods, such as other evaluation systems based on game theory or related equilibrium solutions, to clearly demonstrate the relative performance of this method across different evaluation frameworks?

**Questions:**

In Weaknesses

---

> ### Author Response · Authors · 2024-11-18
>
> We thank the reviewer rQkm for recognising the importance, novelty and careful design of our work as well as providing constructive feedback.
>
> **> Q1: LLE is also an important concept, yet it is only mentioned in section 3.2, causing some logical confusion.**
>
> This is an excellent remark and we apologise for its omission. Indeed LLE is a solution concept that refines the more popularized Nash equilibrium and a key element behind the validity of our approach. LLE is unique, has connections to the Nobel prize winning notion of risk-dominance and benefits from a rich body of literature in psychological studies and behavioural sciences that supports its intuitive nature.
>
> We have moved its discussion earlier to the Background section and we hope this would help readers follow the design choices made around LLE in our method sections.
>
> **> Q2:... "p" represent in the context of Affinity Entropy… needs more explanation**
>
> The hyper-parameter p, sometimes called the *entropic-index*, controls the shape of the Tsallis entropy function, most importantly the steepness of the gradient of entropy near pure strategies. As p approaches zero, Shannon entropy is recovered and the gradient becomes infinitely steep at the boundaries. Note that we have updated our definition of Affinity Entropy with a small modification; it is now concave for all nonnegative p.

---

> ### Author Response · Authors · 2024-11-18
>
> **> Q3:... Could it be beneficial to consider including more benchmark comparison methods, such as other evaluation systems based on game theory or related equilibrium solutions**
>
> We have indeed thought about several other baselines but we concluded that Elo remains the most relevant method to compare against due to its widespread use. We have therefore designed our experiments around different failure modes of Elo, owing to its sensitivity to redundancy and the limited information it reflects in terms of the “game dynamics” in the evaluation data (i.e. as opposed to the equilibrium structure underlying our proposed ratings, which we have exploited for Figure 5).
>
> Other possible baselines we have considered include:
>
> * **Nash averaging:** [1] proposed to use expected payoffs of each action at the max-entropy NE (MENE) but MENE is only unique in two-player zero-sum games. We did not compare to [1] as computing MENE is difficult in n-player general-sum games and would not be unique (see Appendix E);
>
> * **(C)CE-ratings:** [2] proposed computing action ratings at the ME(C)CE of the game. This approach is akin to the setting of CCE(-a) as shown in Figure 3, which we showed to be affected by redundancy. Our max-affinity-entropy criterion provides a practical CCE rating method that remains invariant. Using (C)CE for ratings as a direct alternative to Elo is also open to debate, as it can assign high ratings to **specialist** models that excel at a (small) subset of the prompts. By contrast, both Elo and NE ratings tend to highlight strong generalists;
>
> * **LLE-ratings [2]:** the authors of [2] further hinted at, but did not experiment with the possibility of using LLE for computing a set of NE ratings (recall that LLE defines a unique NE at the limit).
>
> * **Social Choice Theory [3]:** there has been recent interest in using social choice methods for evaluation. However, the invariance to redundancy is usually given among candidates (i.e. models, if we consider prompts to be individual votes) whereas our methods provide invariance to redundancy for all players. SCT methods, as in the case of Elo scores, also do not provide additional insights regarding the importance of prompts;
>
> * **AlphaRank [4]:** in theory, AlphaRank could be a reasonable alternative but we find it challenging to scale in practice due to its excessive memory requirements. Specifically, the transition matrix is quadratic in the number of joint actions ((500 x 17 x 17)^2 elements for the size of the game we study).
>
> Overall, the current practice in the evaluation of LLMs is heavily centred around the Elo rating system and we believe a careful discussion of the practical limitations and failure models of Elo would be a timely contribution to the community.
>
> A good rating system should scale to practical problems and should be non-trivial to manipulate. As we have shown in Figure 1, improvement against an equilibrium rating system leads to well-rounded improvement. In Figure 3, we have shown that our rating system is invariant to exact (and in Supp. Figure 11, near) redundancy. In Figure 5, we have shown that our approach provides an interpretable view of the comparative (dis-)advantages of different models. To the best of our knowledge, our proposal is the first to satisfy these desiderata simultaneously.
>
> ---
>
> [1] David Balduzzi, Karl Tuyls, Julien Perolat, and Thore Graepel. Re-evaluating evaluation. Advances
> in Neural Information Processing Systems, 31, 2018.
>
> [2] Luke Marris, Marc Lanctot, Ian Gemp, Shayegan Omidshafiei, Stephen McAleer, Jerome Connor,
> Karl Tuyls, and Thore Graepel. Game theoretic rating in n-player general-sum games with equilibria, 2022b. URL https://arxiv.org/abs/2210.02205.
>
> [3] Marc Lanctot, Kate Larson, Yoram Bachrach, Luke Marris, Zun Li, Avishkar Bhoopchand, Thomas
> Anthony, Brian Tanner, and Anna Koop. Evaluating agents using social choice theory. arXiv
> preprint arXiv:2312.03121, 2023.
>
> [4] Omidshafiei, Shayegan, et al. "α-rank: Multi-agent evaluation by evolution." Scientific reports 9.1 (2019): 9937.

---

> > ### Comment · Reviewer_itPT · 2024-12-02
> > **Comment**
> >
> > Thank you for your detailed responses.

---

### Official Review · Reviewer_rQkm · 2024-11-05

**Soundness:** 3
**Presentation:** 3
**Contribution:** 3
**Rating:** 6
**Confidence:** 2

**Summary:**

This work studies an equilibrium rating framework that can scale to real-world LLM evaluation and move beyond traditional Elo ratings. The authors start by displaying that Elo ratings are sensitive to redundancy. Then, the authors introduce their specific method, describing the proposed rating method in terms of gamification. The authors then provide several results, finding that the proposed equilibrium ranking framework is fairly consistent with Elo rankings and that the proposed system is invariant to redundancy.

**Strengths:**

Strengths:
+ This paper has an abundance of citations and explains how the proposed work relates to those prior well.
+ There are several interesting, important results. Figure 2, specifically, provides a very clear indication of the utility of the proposed equilibrium ranking approach.

**Weaknesses:**

Weaknesses
- I found the paper a bit difficult to read. As the paper is a bit outside my exact expertise, this difficulty may be attributed to that. However, if other reviewers also found difficulty in understanding the exact details in the methodology section, I would recommend expanding this section with additional intuitive explanations and also expanding some of the descriptions in the appendix.
- Figure captions could be improved by concluding with important takeaways.

**Questions:**

Please address the weaknesses noted above.

---

> ### Author Response · Authors · 2024-11-18
>
> We thank the reviewer rQkm for recognizing the relevance of our work and for their constructive criticism.
>
> **> Q1: … expanding this section with additional intuitive explanations and also expanding some of the descriptions in the appendix.**
>
> This is an excellent remark and we have now included an additional analysis using a toy game of rock-rock-paper-scissors to help build intuition in Appendix D. We have also included in the Background section a definition of the LLE solution concept to better set the stage for the rest of the method section.
>
> The technical complexities of our method primarily reside in the definition around different game-theoretic solution concepts, equilibrium computation and their selection. While most techniques employed in our rating methods have been discussed in prior works in game theory [1, 2], the definition of the affinity entropy is new and bridges the gap between game-theory and in-the-wild LLM evaluation.
>
> Appendix E has an example Chicken game where the equilibrium that is of max-entropy would be different from the one with max-affinity-entropy. We hope this helps provide intuition as to how similarity kernels would react to redundancies via a concrete example.
>
> **> Q2: Figure captions could be improved by concluding with important takeaways.**
>
> We agree with the reviewer and added brief concluding summaries to figure captions where appropriate.
>
> ---
>
> [1] David Balduzzi, Karl Tuyls, Julien Perolat, and Thore Graepel. Re-evaluating evaluation. Advances
> in Neural Information Processing Systems, 31, 2018.
>
> [2] Luke Marris, Marc Lanctot, Ian Gemp, Shayegan Omidshafiei, Stephen McAleer, Jerome Connor,
> Karl Tuyls, and Thore Graepel. Game theoretic rating in n-player general-sum games with equilibria, 2022b. URL https://arxiv.org/abs/2210.02205.

---

> ### Author Response · Authors · 2024-12-02
>
> With the discussion period coming to an end, please do let us know if our responses and revision to the main text have adequately answered your earlier questions or if we can help clarify any further. Thank you!

---

### Official Review · Reviewer_v6QS · 2024-11-08

**Soundness:** 3
**Presentation:** 3
**Contribution:** 3
**Rating:** 8
**Confidence:** 4

**Summary:**

This paper proposes a methodology for rating-based evaluation of Large Language Models (LLMs). The motivation arises from the shortcomings of the current Elo rating system, which may suffer from redundant or invariant prompts, leading to biased or less robust evaluation results. The proposed solution extends the evaluation method from a common 2-player game scenario to a 3-player game, incorporating a measure of separability based on game outcome distributions to favor prompts that better differentiate between models. The quality of the proposed method is assessed using Shannon entropy over model or prompt strategies, and the strategy derivation is based on Coarse Correlated Equilibrium (CCE), which is more accessible compared to the commonly used Nash Equilibrium (NE). Experimental results show that the proposed evaluation method effectively resists the impact of redundant prompts while maintaining similar performance in standard scenarios.

**Strengths:**

* The study of LLM evaluation addresses a promising and urgent need. The proposed method appears to be a reasonable approach for focusing on separability among LLMs, especially for those fine-tuned from the same foundational model.
* Including the condition of prompts in the rating evaluation is a straightforward approach, and I believe this direction will soon gain more popularity.
* The overall narrative is promising, supported by sufficient experiments that demonstrate the robustness of the evaluation method against redundancy.
* Although I did not go into the details in the appendix to verify all theoretical aspects, my rough review suggests that they are likely correct.

**Weaknesses:**

* The main idea is based on the fact that the importance of prompts in average cases may not be balanced; however, the proposed solution may lead to exploitation by niche prompts if LLMs share similar general abilities and some major prompts are marked as redundant. This approach lacks a mechanism to verify prompt redundancy. For a more formal evaluation, it would be better to provide some information on redundancy rather than directly applying the inferred results.
* I found the use case of the proposed method highly related to game balance analysis. It would be beneficial to discuss the strengths and weaknesses of Elo ratings in greater detail, such as their scalar strength representation, which is better than vanilla win values but cannot handle scenarios like Rock-Paper-Scissors. The discussion mainly focused on Elo’s inability to handle redundancy, but it inherently has some resistance to redundancy, at least compared to vanilla win values.
* I have concerns about the focus on the utility of separability. Is high separability always beneficial for evaluation? In my understanding, it can sometimes lead to biased positioning, and it requires more careful assessment rather than direct application.

**Questions:**

* How can we identify redundant prompts or models? Would it be necessary to include human verification of redundancy?

* Scalar rating evaluations still suffer from intransitivity issues, as seen in Rock-Paper-Scissors scenarios. Does LLM evaluation face this problem? If so, does your method have the ability to handle it?

* I found some recent game balance research related to this problem. Is LLM evaluation similar to game balance analysis? For example:
    * Bilevel Entropy-based Mechanism Design for Balancing Meta in Video Games. Adaptive Agents and Multi-Agent Systems (AAMAS), 2023.
        * Discusses game design aimed at maximizing strategy entropy to achieve balance.
    * Identifying and Clustering Counter Relationships of Team Compositions in PvP Games for Efficient Balance Analysis. Transactions on Machine Learning Research (TMLR), 2024.
        * Discusses redundant settings and the problem of using strategy entropy, proposing conditional analysis with evaluation ratings.

* What are the trade-offs of the proposed evaluation method? It is clearly stated that it can alleviate redundancy, but what are the potential drawbacks? Are there risks in excluding some common evaluation prompts, even if they seem redundant? It would be valuable to know if the proposed method has any specific limitations when applied, especially in cases where common prompts might still hold evaluative value despite redundancy concerns.

---

> ### Author Response · Authors · 2024-11-18
>
> We thank reviewer v6QS for recognizing the importance and urgency of the problem we tackle.
>
> **> Q1: … especially for those fine-tuned from the same foundational model.**
>
> In particular, we appreciate this insight which we should highlight more in our writing as there are (increasingly) many models that are fine tuned from the same foundational models (e.g., Llama models) --- performing well against a specific, large family of redundant models would be one way to inflate redundancy-sensitive ratings such as the Elo scores.
>
> **> Q2: …  it would be better to provide some information on redundancy rather than directly applying the inferred results.**
>
> We agree with the reviewer. Indeed, we have updated Figure 4, which now visualises a game instance with artificially introduced redundant prompts, demonstrating how redundant actions would be handled. We observe that redundant actions would split their probability mass at the equilibrium we select, explaining why redundant adversarial prompts had no effect on equilibrium ratings as shown in Figure 3.

---

> ### Author Response · Authors · 2024-11-18
>
> We would like to clarify that although we focused our presentation on the specific use-case of side-by-side LLM comparison, our proposed rating framework is easily generalised in several ways.
>
> **> Q3: … concerns about the focus on the utility of separability.**
>
> On gamification: the game design (i.e. utility functions, number of players, …) depends on the applications and the behaviors we would like to see at an equilibrium. The separability utility function we chose would lead to model ratings that focus on performance on prompts discriminative among strong models. Other game designs could be equally valid and valuable as the reviewer pointed out.
>
> **> Q4: … Coarse Correlated Equilibrium (CCE), which is more accessible compared to the commonly used Nash Equilibrium (NE)**
>
> On solution concept: which solution concept to use depends on downstream applications. If we want the model ratings to reflect strengths of a model in any *strategic niche*, then the CCE solution concept could be preferred. If we want model ratings to reflect how generally capable each model is, then NE ratings would make more sense. Our rating framework leaves room for practitioners to lean one way or another depending on applications.
>
> **> Q5:  How can we identify redundant prompts or models? Would it be necessary to include human verification of redundancy?**
>
> On redundancy: how we define redundant prompts or models is flexible, and in fact, this flexibility to re-interpretation of redundancy is precisely a strength we designed for in desired property P4 (section 3.3). In experiments, we chose to call two prompts similar if they rank all model pairs identically (see distance matrix D in equation 5). However, for example, distance matrix D in equation 5 could instead be defined to be the expected squared difference between payoffs **+** the edit-distance between the two prompt strings or cosine-distance between their embeddings.
>
> To clarify this distinction further, we are not arguing one way or another how to correctly measure prompt redundancy. It would make sense to verify a definition of prompt similarity with human raters prior to passing this definition to our method. We are proposing a game-theoretic technique that rates models/prompts in a way that respects a user-provided definition of redundancy (and also approximate redundancy). In that setting, our proposed approach is uniquely capable of handling redundancy.
>
> **> Q6: … Scalar rating evaluations still suffer from intransitivity issues, as seen in Rock-Paper-Scissors scenarios.**
>
> Our equilibrium ratings account for intransitivity and we could indeed make it clearer in writing. We have now introduced a new visualisation of our rating method applied to the **Rock-Rock**-Paper-Scissors game in Appendix D. All four actions: Rock, Rock, Paper, Scissors would receive identical ratings under our rating framework, with an interpretable contribution plot (similar to Figure 5) *explaining* their ratings. For instance, we see the paper action receives a rating of 0 because if it were to play paper instead of its equilibrium distribution (uniform over R/P/S), it would lose and win in equal proportion against scissors and the two rocks respectively.
>
> **> Q7: Are there risks in excluding some common evaluation prompts, even if they seem redundant?**
>
> This is a great question. There are many equilibria in real-world evaluation games that we construct. Indeed, we have empirically computed 128 Nash equilibria in Supplementary Figure 6. For CCE, there’s typically a continuous polytope of (infinitely many) equilibria.
>
> As such, which equilibrium we select matters and some of these equilibria would indeed unduly ignore “common prompts” that, for example, rank the overall stronger models high and weaker models low. However, these do not tend to occur with our selection procedure. For the NE solver, we propose to select the LLE of the game which tends to be risk-dominant. For CCE we propose to select the max-affinity-entropy CCE. Both procedures tend towards highly-mixed prompt distributions, if such an equilibrium exists.
>
> In other words, our method is robust against (exact or near) redundancy, but would not unduly ignore “common prompts”. The degree to which two prompts are considered similar is also controlled by the users via the similarity kernel: using a small RBF kernel variance would guard against redundancy in the most strict sense, which is what we did in experiments (we used a kernel variance of 1e-6, see Supp. Section E.2).
>
> **> Q8: Related works on game balance research.**
>
> Indeed this line of work becomes relevant once we consider LMSYS evaluation in the context of game design. The equilibrium prompt distribution would in effect re-adjust the prompt distribution such that no single topic overshadows others. This is similar to the pattern shown in the "NE prompts" column in Figure 2. We have revised our Conclusion section accordingly.

---

> > ### Comment · Reviewer_v6QS · 2024-11-22
> >
> > Thank you for your response. I have some questions:
> >
> > 1. Is there a typo in line 1155 (appendix)?
> >
> >
> > 2. Regarding the intransitivity of scalar ratings, do you mean that although scalar ratings cannot fully represent intransitivity, showing the contribution of each rating could provide additional explanations for this phenomenon?
> >
> >
> > 3. About the rating mechanism: since the selection of prompts is based on separability, which is influenced by the LLMs being evaluated, is there any clarification in the main paper? For example, if there is a strong reasoning question in the evaluation prompt pool that humans consider valuable, but all current models perform poorly on it, such prompts may not contribute to separability and might be discarded. Would this lead to overlooking a future model with strong reasoning capabilities because the relevant prompts have already been removed?
> >
> >
> > 4. On the foundation of using entropy: the game balance paper "Identifying and Clustering Counter Relationships of Team Compositions in PvP Games for Efficient Balance Analysis" discussed how maximizing the entropy of strategies may not always align with achieving true "balance." In the context of LLM evaluation, this would be analogous to general ability. They proposed counting how many settings are non-dominated by others to quantify the diversity of dynamics. In your NE-based rating system, is there any metric to quantify the diversity or capacity of the evaluation index? For instance, the number of prompts retained after filtering? It seems important to provide information about the prompt pool's diversity or domain/task coverage, as a more diverse and comprehensive prompt pool would likely yield more convincing rating results.

---

> > > ### Author Response · Authors · 2024-11-22
> > >
> > > > Is there a typo in line 1155 (appendix)?
> > >
> > > We reviewed this sentence but did not spot an obvious typo in the line that reads “Given that prior work has shown…”. However, we have rephrased the section for clarity. Please let us know if it still doesn't read well.
> > >
> > > > scalar ratings cannot fully represent intransitivity...
> > >
> > > The reviewer is right that scalar ratings cannot define strategic cycles: given the zero ratings assigned to R, P and S, we cannot tell if there is a cycle, or in which direction a cycle should form.
> > >
> > > However, equilibrium ratings *account for* strategy cycles --- in a game of RRPS, our proposed equilibrium ratings show them to be of *equal strength* because the underlying equilibrium structure *recognises* that they are in a cycle and no one action should be better than the others. Contrast this with the Elo ratings: in a game of RRPS, the paper action would receive a higher Elo rating because updates to Elo ratings are always made by pairwise comparisons.
> > >
> > > >  if there is a strong reasoning question in the evaluation prompt pool that humans consider valuable, but all current models perform poorly on it, such prompts may not contribute to separability and might be discarded...
> > >
> > > This is a great question. This is exactly why we do *not* suggest discarding prompts that are not played at an equilibrium at one moment in time --- a prompt can be *too hard* now but becomes highly discriminative once a first model starts to solve it, at which point we expect such a prompt to positively contribute to this trailblazing model's rating!
> > >
> > > The fact that the equilibrium prompt distribution adapts to the skill levels demonstrated by the pool of models is a strength --- the influence of each prompt depends on how much it helps us distinguish between model candidates.
> > >
> > > The question of "which prompts can be removed and which one cannot" deserves its own dedicated investigation which is why we avoided any speculation in this work.

---

> > > > ### Author Response · Authors · 2024-11-22
> > > >
> > > > >  It seems important to provide information about the prompt pool's diversity or domain/task coverage, as a more diverse and comprehensive prompt pool would likely yield more convincing rating results. is there any metric to quantify the diversity or capacity of the evaluation index? For instance, the number of prompts retained after filtering?
> > > >
> > > > We have indeed discussed if we can translate equilibrium strategies into some metrics on diversity. The main difficulty, however, is that we do not have ground truth labels of precisely which topic each prompt in the Arena Hard 500 dataset investigates. We attempted at using an off-the-shelf LLM to assign cluster labels to each prompt but the result turned out highly biased --- a vast majority of these prompts are related to coding or computer system admin topics. Much fewer prompts target creative writing, language tasks, etc.
> > > >
> > > > In fact we wonder if we were observing a similar effect as demonstrated in the Figure 1 / "Additional Prompts" setting where if we prioritise prompts that separate models *on average* [1], then the prompt diversity may collapse onto one skill out of many.
> > > >
> > > > > maximising the entropy of strategies may not always align with achieving true "balance." ...
> > > >
> > > > We agree with the reviewer. Entropy-maximisation is a reasonable general principle but it does not imply any specific notion of "balance". This is especially true when actions can be redundant (e.g. RRRPS).
> > > >
> > > > In game theory, there has been long standing debates on which equilibrium (out of many) one should choose. The most notable selection criteria here is that of risk-dominance [2] (indeed not necessarily entropy-maximising). A key reason for its prominence is that many psychological studies showed that humans tend to prefer risk-dominant equilibria too. Unfortunately, the notion of risk-dominance is most clearly discussed in two-player two-action games (e.g. a game of Stag Hunt) and we could not make a serious main text discussion in our game which is larger and much more general.
> > > >
> > > > Nevertheless, in Appendix F.4, we made an attempt at showing that the LLE our procedure selects is empirically risk-dominant. Indeed, we empirically computed 128 mixed-strategy NEs for the 3-player LLM evaluation game and showed that the LLE we chose is the least risky in case of mis-coordination --- demonstrating empirical risk-dominance. More interestingly, we observe the expected payoffs to all 3 players to be quite balanced (Figure 10 (Bottom), notice the height of each coloured bar). In other words, compared to other equilibria, the one our method selects is *fair* for all players. We note that in practice, the LLE tends to be mixed, though not necessarily entropy maximising. We agree this notion of risk dominance or fairness between players can have further connection to the game balance literature too, and we will read this second paper and include reference to this where appropriate.
> > > >
> > > > Finally, we clarify that we developed the notion of affinity-entropy as a tool to steer the equilibrium selection process towards an LLE of the game in a clone-invariant way. It does not imply that the LLE solution we computed would be entropy-maximising.
> > > >
> > > > ---
> > > >
> > > > [1] https://lmsys.org/blog/2024-04-19-arena-hard/
> > > >
> > > > [2] John C Harsanyi and Reinhard Selten.  A general theory of equilibrium selection in games. MITPress Books, 1, 1988.
> > > >
> > > > ---
> > > >
> > > > We thank reviewer v6QS again for engaging with our work thoughtfully and constructively.

---

> > > > > ### Comment · Reviewer_v6QS · 2024-11-29
> > > > >
> > > > > Thank you for your explanation. Including these discussions in the final revision, even if only in the appendix, would be very helpful for future applications, especially in clarifying the limitations and assumptions. Assuming such discussions are added in the final version, I have adjusted my score to reflect this potential improvement: 8 for a final revision with more detailed discussion, or 6 for the current version.

---

> > > > > > ### Author Response · Authors · 2024-11-29
> > > > > >
> > > > > > We thank reviewer v6QS for their engagement and constructive feedback throughout the discussion period.
> > > > > >
> > > > > > While we cannot upload new revision now, we will include in the Appendix section two discussions:
> > > > > >
> > > > > > 1) a topic-clustering analysis of the ArenaHard dataset with an off-the-shelf LLM, complemented with a discussion on how the prompt-by-topic distribution in the data differs from the prompt-by-topic distribution at an equilibrium (from which ratings are computed);
> > > > > >
> > > > > > 2) we will expand our discussion on the notion of risk-dominance (Appendix F.4) to relate to game balance literature and also show in an updated Figure 10 that the entropy of each of the 128 NEs do not correlate with their risk-dominance (although the risk-dominant LLE we compute is among the solutions with higher entropies, many NEs of high entropy can be quite *risky*).

---

### Author Response · Authors · 2024-11-18

Thank you to all the reviewers for taking the time to read our paper and provide helpful feedback. We are encouraged by your words!

We have already made several changes to the paper (highlighted in red) and we have uploaded a new revision to the main text as well as the supplementary material. These include improving the discussion of equilibrium concepts in the Background section, more informative captions, an improved affinity entropy definition, improved Figure 4, simplified Figure 5, as well as intuitive warm-up rating examples in Appendix D and E.

We will continue to incorporate your feedback and improve the paper throughout the discussion period and we look forward to further discussions.

**On the positioning of our paper**: our primary goal is to draw attention to issues that can arise when researchers and practitioners rely on redundancy-sensitive rating methods to measure progress as a community. We then propose a game-theoretic rating method that can scale, is intuitive, clone-invariant and captures intransitivity. Our hope is that this rating method can be of interest to LLM evaluation system designers (e.g. LMSYS, Alpaca) as we believe this is an area that deserves care given its relevance.

This is not to say that our method is limited to LLM evaluation but rather LLM evaluation is to us the most influential open-ended evaluation system in the community. As such, we have primarily surveyed rating method literature (for comparison), game theory literature (as our method relies on them) and omitted several relevant pieces of literature in LLM evaluation which we are including in revision. We thank all reviewers for their feedback which has helped made our paper more thorough and accessible.

---

### Author Response · Authors · 2024-11-25

With the end of the discussion period fast approaching, please do let us know if our comments and revision have adequately addressed your concerns and if you have any questions outstanding.

We hope our work can start a timely discussion regarding in-the-wild LLM evaluation in the broader community and we sincerely thank all reviewers again for taking the time to review our work and for their help in improving our manuscript.

---

### Meta-Review · Area_Chair_3Rpx · 2024-12-23

**Metareview:**

Cool idea. Novel. Good paper. Deserves a spotlight.

**Additional Comments On Reviewer Discussion:**

Good discussion.

---

### Decision · Program_Chairs · 2025-01-22

Accept (Poster)